

# Long-term river trajectories to enhance restoration efficiency and sustainability on the Upper Rhine: an interdisciplinary study (Rohrschollen Island, France)

David Eschbach [1], Laurent Schmitt [1], Gwenaël Imfeld [2], Jan-Hendrik May [3,4], Sylvain Payraudeau [2], Frank
Preusser [3], Mareike Trauerstein [5], Grzegorz Skupinski [1]

[1] Laboratoire Image, Ville, Environnement (LIVE UMR 7362), Université de Strasbourg, CNRS, ENGEES, ZAEU LTER, Strasbourg, France
[2] Laboratoire d'Hydrologie et de Géochimie de Strasbourg (LHyGeS UMR 7517), Université de Strasbourg, CNRS, ENGEES, Strasbourg, France
[3] Institute of Earth and Environmental Sciences, University of Freiburg, Freiburg, Germany
[4] Present address: School of Geography, University of Melbourne, Australia
[5] Institute of Geography, University of Bern, Bern, Switzerland

*Correspondence to:* David Eschbach (david.eschbach@live-cnrs.unistra.fr)

**Abstract.** While the history of a fluvial hydrosystem can provide essential knowledge on present functioning, historical context remains rarely considered in river restoration. Here we show the relevance of an interdisciplinary study for improving restoration within the framework of a European LIFE+ project on the French side of the Upper-Rhine (Rohrschollen Island). Planimetric evolution combined with historical high flow data enabled to reconstruct pre-disturbance hydromorphological functioning and major changes that occurred on the reach. A deposition frequency assessment combining vertical evolution of the Rhine thalweg, chronology of deposits in the floodplain and a hydrological model revealed that the period of vertical incision in the main channel corresponded to high rates of narrowing and lateral channel filling. The analysis of filling processes by Passega diagram and IRLS dating highlight that periods of engineering works were closely related to fine sediment deposition, which present also concomitant heavy metal accumulation. In fact, current fluvial forms, processes and sediment chemistry around the Rohrschollen Island directly reflect the disturbances that occurred during past correction works, and up to today. Our results underscore the advantage of combining functional restoration with detailed knowledge of past-trajectory to: (i) understand the functioning of the hydrosystem prior anthropogenic disturbances, (ii) characterize the human-driven morphodynamic adjustments during the two last centuries, (iii) characterize physio-chemical sediment properties to trace anthropogenic activities and evaluate the potential impact of the restoration on pollutant remobilization, (iv) deduce post-restoration evolution and (v) evaluate efficiency and sustainability of the restoration effects. We anticipate our approach to expand the toolbox of decision-makers and help orientating functional restoration actions in the future.

## 1 Introduction

During the two last centuries, numerous engineering works (e.g. channelization or damming), aimed at flood control, navigation improvement, expansion of agriculture or hydropower production, have altered the functioning of European large



rivers (Brookes, 1988; Kondolf and Larson, 1995; Petts et al., 1989), including aquatic and riparian habitats and biodiversity (Amoros and Petts, 1993; Bravard et al., 1986; Dynesius and Nilsson, 1994). In order to balance these impacts by recovering fluvial processes (Bravard et al., 1986; Hering et al., 2015; Naiman et al., 1993, 1988) and ecosystem services (Acuña et al., 2013; Large and Gilvear, 2015; Loomis et al., 2000), an increasing number of restoration projects have been carried out over
the last decades (Kondolf and Micheli, 1995; Wohl et al., 2005). In Europe, this trend has been supported by the Water Framework Directive (IKSR-CIPR-ICBR, 2005; WFD, 2000). Restoration activities progressively target hydromorphological processes and functioning rather than ±staticø fluvial forms (Arnaud et al., 2015; Beechie et al., 2010; Jenkinson et al., 2006). Numerous studies have shown that current river functioning results from complex long-term trajectories driven by natural and anthropogenic factors at different spatio-temporal scales (e.g. Bravard and Magny, 2002; Brown, 1997; Gregory and Benito,
2003; Starkel et al., 2006; Ziliani and Surian, 2012). These trajectories provide a relevant basis to infer future trends and management principles (Sear and Arnell, 2006; Bravard, 2003; Fryirs et al., 2012). In order to understand the complete range of functional changes, a comprehensive understanding of human-driven channel adjustments over the last centuries therefore appears crucial in river restoration. Despite significant research efforts over the last two decades, however, integrating pluri-secular temporal trajectories into restoration projects remain an exception (Fryirs et al., 2012; Sear and Arnell, 2006).

In large modified hydrosystems such as the Upper Rhine River, current lateral extent of the floodplain result from past disturbances that occurred during engineering works (Herget et al., 2005, 2007). Hydromorphological dynamics, chemical pollutions and depositional processes were strongly impacted by diking along the main channel and disconnecting of lateral channels since the beginning of the 19[th] century (Tümmers, 1999). Several studies have focused on the storage and remobilization of heavy metals (Ciszewski and Gryar, 2016; Falkowska et al., 2016; Grygar and Popelka, 2016; Schulz-Zunkel
and Krueger, 2009) and/or organic pollutants in major floodplains worldwide (Berger and Schwarzbauer, 2016; Zimmer et al., 2010; Lair et al., 2009). Most studies concerned with the Rhine focused on the industrializes Lower Rhine region including the Rhine-Meuse Delta (de Boer et al., 2010; Evers et al., 1988; Goth et al., 2001; Middelkoop, 2000). In comparison, the Upper Rhine region is two times less contaminated than lower part of the Rhine with respects to total concentrations of both polychlorinated dibenzo(p)dioxins (PCDDs) and polychlorinated dibenzofurans (PCDFs) in sediments and tracers of industrial
activities (Evers et al., 1988). However, contamination histories are still not well known and reference studies considering functioning or disturbance histories are missing.

Studying past functioning and disturbance histories as keys to understanding current forms and processes of floodplains can provide insights into current functioning and hydromorphological sensitivity to changes (Kondolf and Larson, 1995; Mika et al., 2010). Historical hydromorphological adjustments should be evaluated on an accurate spatial scale and a high temporal
resolution to identify past evolutionary processes and causal relationships (Bogen et al., 1992; Horowitz et al., 1999). Interdisciplinary and retrospective studies, however, rarely combine different data sources to obtain a comprehensive view of functional changes (Bravard and Bethemont, 1989; Gurnell et al., 2003; James et al., 2009; Lawler, 1993; Lespez et al., 2015; Rinaldi et al., 2013; Trimble and Cooke, 1991). Furthermore, historical studies rarely consider sediment dating and pollution (Bogen et al., 1992; Garban et al., 1996; Horowitz et al., 1999; Woitke et al., 2003).



Within the framework of a functional restoration project on the Rohrschollen Island, we embarked for an interdisciplinary and retrospective pluri-secular study to provide a holistic understanding of the functional temporal trajectory of the fluvial hydrosystem, rather than to determine reference states. We hypothesized that long-term temporal trajectories allow to identify

driving-factors, amplitude and response time of disturbances, and to assess potential benefits and limits of the restoration. This approach bears potential to accompany actions of functional restoration in order to maximize efficiency and sustainability, and infer future evolutionary trends. To test these hypotheses, this study combines horizontal (planimetric) and vertical (thalweg evolution, filling chronology) dynamics to: (i) understand the functioning of the hydrosystem prior anthropogenic disturbances, (ii) characterize the human-driven morphodynamic adjustments during the two last centuries including sediment transport and

deposition processes as well as geochronology, (iii) characterize physio-chemical sediment properties (e.g. heavy metals and organic contaminant concentrations) to trace anthropogenic activities and evaluate the potential impact of the restoration on pollutant remobilization (Fedorenkova et al., 2013; IKSR-CIPR-ICBR, 2014; Middelkoop, 2000), (iv) deduce post-restoration evolution and (v) evaluate efficiency and sustainability of the restoration effects (Grabowski and Gurnell, 2016; Sear et al., 1994).

## 15  2   Study area

With a total length of 1,250 km and a drainage basin of about 185,000 km², the Rhine is the third largest river of Europe. Located between Basel and Bingen (Fig. 1-a), the Upper Rhine Graben is 35-50 km wide and 310 km long. Hydrology in the southern part of this sector is characterized by a nivo-glacial regime and a mean discharge of 1,059 $m^3.s^{-1}$ (1891-2011; Basel gauging station; Uehlinger et al., 2009). Slope decrease and inherited geomorphological factors explain the longitudinal

evolution of the channel pattern (Carbiener R., 1983; Schmitt et al., 2016; Fig. 1-b). Since the middle of the 19th century, three successive engineering works modified drastically the hydrosystem: (i) the correction stabilized the main channel between two artificial banks and the floodplain between two high flow dikes, (ii) the regularization consisted to build alternative in-channel groyne fields to improve navigation and (iii) the canalization by-passed the corrected main channel southern Strasbourg. Nowadays, the river consists of a single channel which is locally by-passed by artificial canalised sections. Northern

Strasbourg, the canalization concreted the Rhine bed itself.





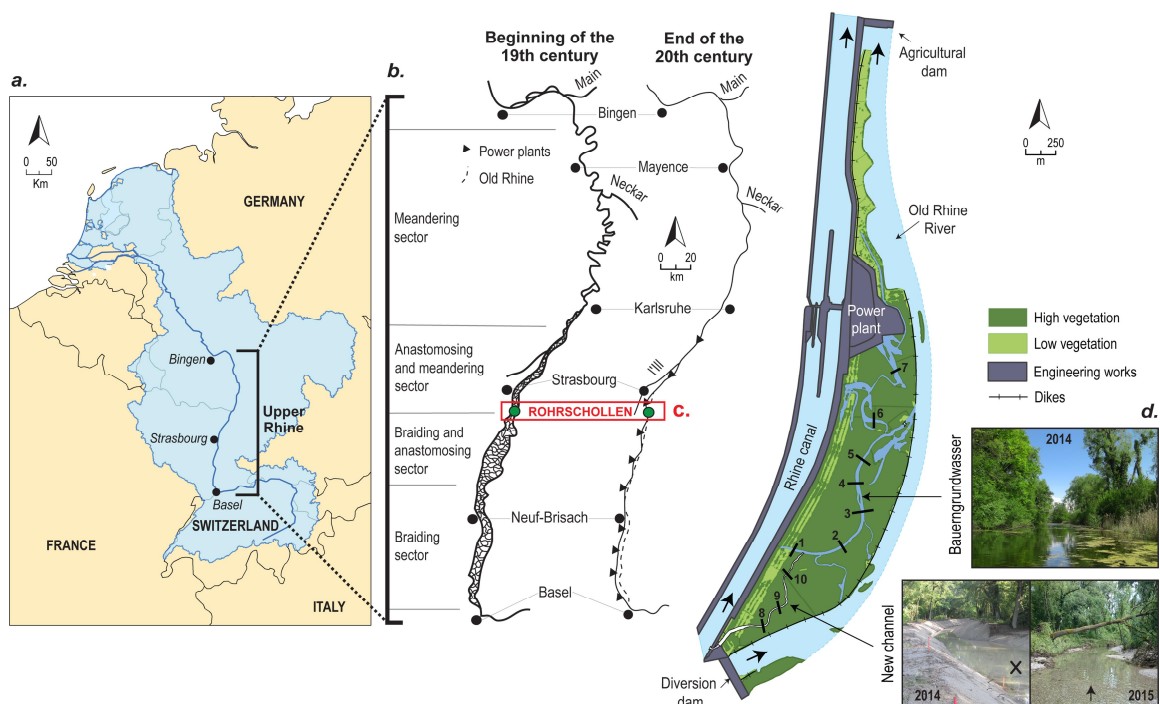

**Figure 1: (a) Location of the Upper Rhine Graben, (b) channel pattern sectorization and evolution from the 18ᵗʰ century to present (Schmitt et al., 2009), (c) location of the study site and map of Rohrschollen Island, (d) pictures of the Bauerngrundwasser and evolution of the new channel.**

The Rohrschollen artificial Island is located 8 km South-East of the city of Strasbourg and owes its existence to the construction of a power plant in 1970. The Island is enclosed by the Rhine canal to the west and the Old Rhine River to the east, which corresponds to a by-pass of the corrected Rhine (Fig. 1-c). On the southern part, a diversion dam diverts up to 1,550 $m^3.s^{-1}$ for usage by the power plant. When the discharge is less than 1,563 $m^3.s^{-1}$, an instream discharge of 13 $m^3.s^{-1}$ flows to the Old Rhine. On the northern part of the island, an agricultural dam built in 1984 maintains a constant water level in the Old Rhine

at 140 m NN (NormalNull) in order to increase groundwater level for agricultural purpose. When floods exceed 2,800 $m^3.s^{-1}$ (2-year instantaneous flood), the agricultural dam is raised for flood retention (IKSR-CIPR-ICBR, 2012), but flooding remains static. The island is crossed by an anastomosing channel (Bauerngrundwasser), which is disconnected from the Rhine canal at its upstream extremity and connects to the Old Rhine further downstream (Fig. 1-c & d). Further north an additional minor channel flows towards the Rhine Canal. The water level of the entire length of the Bauerngrundwasser is artificially maintained

by the hydraulic backwater of the agricultural dam (Fig. 3-c). Classified as a natural reserve since 1997, the island has recently been restored (European LIFE+ project) in order to recover typical alluvial processes and biodiversity, including dynamic floods, bedload transport, active morphodynamics, or hygrophilous tree species. In the southern part of the island a large



floodgate was built in 2013 and a new upstream channel of 900 m length was excavated (Fig. 1-c & d). The downstream end of this channel is connected to the Bauerngrundwasser channel. Water input from the flood gate ranges between 2 m$^3$.s$^{-1}$ (when Q Rhine < 1550 m$^3$.s$^{-1}$) and 80 m$^3$.s$^{-1}$ (when Q Rhine > 1550 m$^3$.s$^{-1}$). Our study addresses embedded spatial scales: (i) the entire study site, which corresponds to the fluvial hydrosystem area around the natural reserve (about 1-3 km beyond the perimeter

5    of the later), (ii) the Rohrschollen Island which corresponds to the natural reserve area, (iii) seven transects on the Bauerngrundwasser used to characterize sediment transport and depositional processes, and (iv) two sediment pits excavated near the new and the old channels (Fig. 4-a, 2010) in order to date sediment deposition and assess sediment pollution.

## 3    Material and methods

In this study, we have adopted an interdisciplinary approach that combines hydrological retrospective modelling with

10    limnimetric, topographic (levelling and DEM) and hydrogeologic data as well as data on sediment filling processes, depositional chronology and geochemical characteristics (Fig. 2).

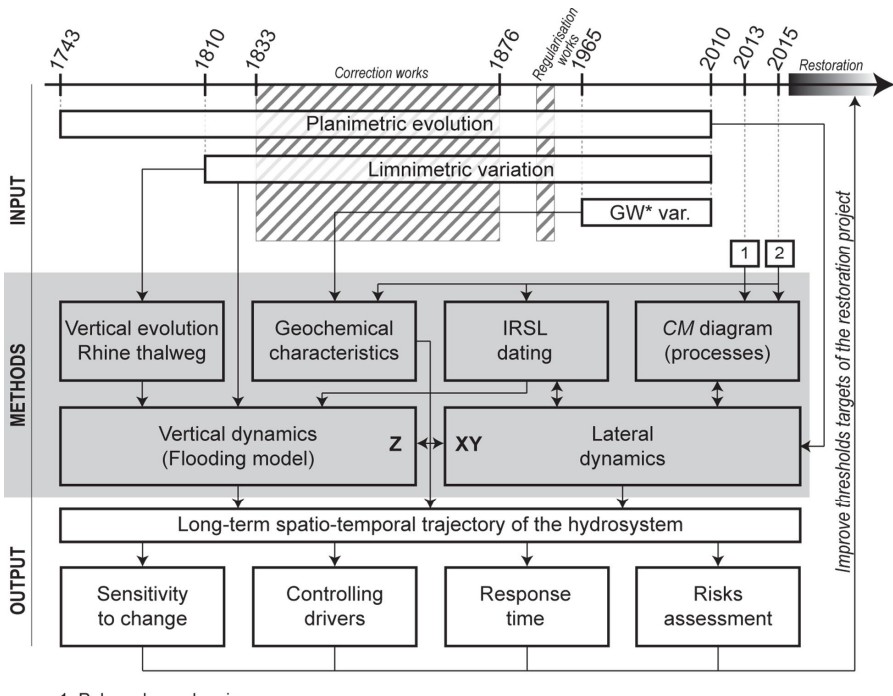

**Figure 2: Methodological approach to analyse long-term spatio-temporal trajectory of the hydrosystem by combining historical multi-source data with *in situ* morpho-sedimentary prospections and geochemical investigations.**



### 3.1 Planimetric analysis

The historical planimetric analysis covers a period of about 260 years (from 1743 to 2010) and was carried out in ArcMap (ESRI v.10.3) using six historical maps and two sets of aerial photographs. The map from 1828 compiled during the demarcation of the Franco-German border was georeferenced on the IGN BD ortho 2007 base map and used as a base layer

for georeferencing of the 1743, 1778, 1838, 1872 and 1926 maps. Aerial photographs were georeferenced using the 2007 orthophotograph as a base layer. Fixed position objects such as churches, road crossings or bank protection structures were used as control points. Between nine and twelve control points were selected for each historical map, and between five and seven for the aerial photographs. Total root mean square error (RMSE) ranged from 0.94 to 25 m and increased with the distortion and the imprecision of the oldest maps, especially the 1743 and 1778 maps (Fig. 4-c). However, the RMSE distortion

is satisfactory considering the inherent relative imprecision of these maps and the difficulty to determine anchor points between old maps and 2007 orthophotography. Aerial photographs were selected at low flow water level to enable comparison between morpho-ecological surfaces and active channel and gravel bar surfaces which are particularly sensitive to discharge variations (Rollet et al., 2014). Morpho-ecological units were then manually digitized at the detailed scale of 1:1000 to 1:2000 based on 8 maps. According to the typology developed by Dufour (2005), four classes and fourteen subclasses were determined. Surface

areas of each class and time slice were calculated to quantify the planimetric evolution. The results of the surface analyses were converted to a percentage ratio to facilitate regional interpretation of the morpho-ecological evolution. Two scales were considered: the study area of the Rohrschollen Island with a total surface of 2,181 ha and the area of the natural reserve with a total surface area of 314 ha.

### 3.2 Analysis of vertical data

#### 3.2.1 Limnimetric and piezometric analysis

The vertical evolution of the Rhine thalweg was studied based on historical limnimetric data and bibliographic references (Bensing, 1966; Bull, 1885; CECR, 1978). Limnimetric data were compiled for low water discharge ($\sim 540$ m$^3$.s$^{-1}$) at the Marlen gauging station (Kilometre Point 295; Felkel, 1969; Jeanpierre, 1968). The piezometric analysis was achieved using a German database (LUBW: Landesanstalt für Umwelt, Messungen und Naturschutz Baden-Württemberg) and the French

regional groundwater database (APRONA, Association pour la PROtection de la Nappe phréatique de la plaine døAlsace), by selecting datasets close to the Rohrschollen reach.

#### 3.2.2 Palaeochannel corings

Seven coring transects were distributed along the Bauerngrundwasser to cover the different morphological characteristics of the study area (e.g. Fig. 3). Six sediment cores per transect were hand-augered on both channel banks to measure the thickness

of the post-correction deposits. From the 42 cores, a total of 81 sediment samples were taken at different depth of the filling layer. Two additional samples were extracted from the channel bottom at transects two and four with a piston sampler.




### 3.2.3 Pit excavations

In addition to the transect-based prospection, two pits were excavated up to the gravel bottom, on the right bank of the two channels. Locations of the pits were determined by (i) identifying main filling sectors revealed by old maps and corings survey, and (ii) the proximity to the potential future erodible banks (concave banks; Fig. 3-a). Stratigraphical Units (SU) were defined

in the field on the basis of colour and textural differences. Two large topographical cross-sections (600 m for the pit 1 and 800 m for the pit 2) intersecting both pits were extracted from the DEM to interpret the dynamics of fine sediment deposition in relation to the initial elevation of each pit, the thickness of the stratigraphical units and the flooding regime (Fig. 3-b).

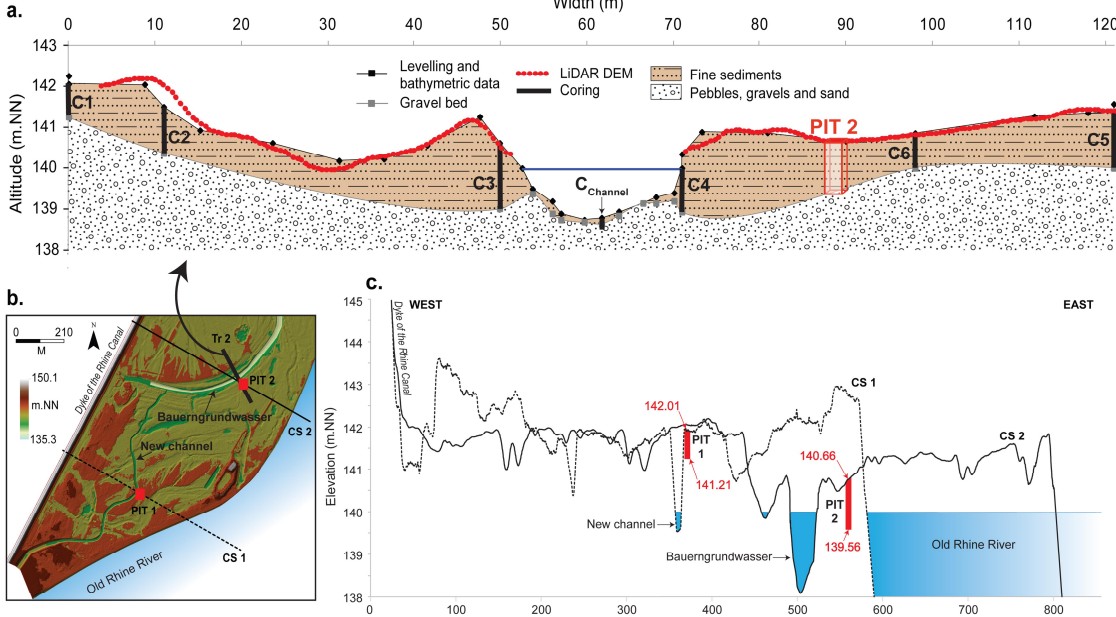

**Figure 3: (a) Example of a transect (Tr 2) based on levelling and bathymetric data. Levelling data are compared to LIDAR DEM**
**data. Location of corings and Pit 2 is also given, (b) location of Tr 2, the two excavation pits (red squares) and topographical profiles crossing the two pits, (c) overlaying of the two topographical profiles.**

## 3.3 Characteristics of the sediments

### 3.3.1 Grain size analysis

Depending on width and thickness of the filling, 31 samples from three coring transects distributed along the entire length of

15 the Bauerngrundwasser (2, 4 & 5 in figure 1) and two samples from the channel bottom (2 & 4 in figure 1) were selected for grain size analysis. In addition, one sample per SU were taken from each excavated pit (Fig. 7). Munsell colour and qualitative SU description were completed in the field. Cumulative grain size distribution and a sorting index were obtained from measurements with a Beckman Coulter laser diffraction particle size analyser. Then, the soil organic carbon ratio was



determined by the loss on ignition method (375°C during 16 hours). To further characterize transport and depositional processes, we used the CM diagram method (Bravard and Peiry, 1999; Passega, 1964, 1977).

### 3.3.2 Geochemical and organic pollutants analyses

The 10 samples from the two pits (Fig. 7) were air-dried at 20 °C and sieved (<2 mm). Dried sediments were pulverized (<63
µm) using an agate disk mill prior to alkaline fusion and total dissolution by acids. Measurement of elemental concentrations was done as described previously (Duplay et al., 2014) by inductively coupled plasma atomic emission spectrometer and mass spectrometer analysis (ICP-AES and ICP-MS) using the geological standards BCR-2 (US Geological Survey, Reston, VA, USA) and SCL-7003 (Analytika, Prague, Czech Republic) for quality control. An enrichment factor (EF) was used to compare changes of Zn, Cr, Ni, Cu, Pb and Cd concentrations in the pit 1 and pit 2 profiles with the reference soil collected in the
deepest SU of the considered pit:

$$EF_{HM} = \frac{\left(\frac{HM_{sample}}{Ti_{sample}}\right)}{\left(\frac{HM_{reference}}{Ti_{reference}}\right)} \qquad (1)$$

where HM and Ti are, respectively, the concentrations of the considered heavy metal and Ti (mg kg$^{-1}$ d.w.) in the SU sample of the pit and the reference soil. Concentrations of heavy metals were normalized relative to titan (Ti) as a conservative element to limit EF variations due to local heterogeneities as it displays a low relative standard deviation (<0.03) over both pits (Reimann and de Caritat, 2005).
Based on previous studies dedicated to the contamination of the Rhine sediments (Fedorenkova et al., 2013; van Helvoort et al., 2007), 38 legacy and modern organic pollutants (including 30 pesticides, the hexachlorobenzene and 7 polychlorinated biphenyls) were analysed by liquid chromatography and gas chromatography mass spectrometry (LC-MS and GC-MS) with quantification limits ranging from 0.015 and 0.05 µg.g$^{-1}$ (see Tables S1).

### 3.3.3 Depositional chronology

Dating of sediments sampled from both pits was carried out using Infrared Stimulated Luminescence (IRSL). The detailed procedures and methodological aspects are discussed in Preusser et al. (2016). In summary, IRSL (stimulated at 50°C) of sand-sized feldspar grains was measured applying both small aliquot (ca. 100 grains) and single-grain techniques. When applying the Minimum Age Model (MAM) on single-grain data sets, the estimated ages coincides with the expected age of the sediment. At the same time, MAM ages calculated for the small aliquot data sets overestimate the known age by up to 200 years (>100
%). This is explained by partial resetting of the IRSL signal prior to deposition and masking effects when measuring several grains at the same time. Presented here are the results previously published by Preusser et al. (2016) from pit 1 together with additional samples taken from pit 2 (Table S1). For the new data, we again observed significant older ages when measuring several grains at the same time for most of the samples. Only for sample IDEX2-107, multiple and single-grain approaches





gave the same result. However, for this sample too small grains (100-150 µm) were measured using Risø single-grain discs. This results in accumulating a few (ca. 5) grains being measured at the same time and likely similar averaging effects as observed for larger aliquots. Hence, we consider these ages as maximum estimates.

### 3.3.4 Flooding frequency assessment

To determine the depositional chronology of the two pits and reconstruct flooding frequency over time, we developed a simple flooding model based on (i) the results of the vertical evolution of the corrected Rhine thalweg close to the Rohrschollen Island, (ii) the thickness of fine sediments at both pits, and (iii) IRSL dating. Elevation of the pits at each time slice was determined by the gradual sediment accumulation and the corresponding elevation increase between the two adjacent IRSL dates. Active channel width in vicinity to the pits was measured from the 1828, 1838 and 1872 maps. Flooding water depth was determined by subtracting the elevation of the pits from the thalweg elevation for each time slice (Fig. 5-c). For the three dates corresponding to the 1828, 1838 and 1872 maps, we determined the maximum discharge using the hydrogram ($Q.max$) at Basel and the limnimetric data at the Kehl bridge which is located about 7 km downstream the study area. We calculated for these dates and the corresponding bankfull discharges the sections ($S$), the mean water slopes ($I$), the hydraulic radius ($Rh$) and the roughness ($k$) close to the pits. This allowed to estimate flood discharges ($Q.flooding$) for both pits during the entire period from 1828 to 1970 (end of the canalization) using the Manning-Strickler equation (Eq. 2):

$$Q.flooding\ pit_x\ at\ date_n = S.date_n \times I^{1/2}.date_n \times Rh^{2/3}.date_n \times k \tag{2}$$

Finally, we compared these results with the limnimetric variations (historical hydrogram) to determine the frequency and the intensity of historical floods in each pit (Fig. S2; Fig. 5-c).

## 4 Results and discussion

### 4.1 Hydromorphological dynamics before the beginning of the correction works (up to ca. 1833)

Analysis of the three earliest historical maps (Fig. 4-a, 1743, 1778 and 1828) at the scale of the entire study site documents the natural morphodynamics along the Upper Rhine before the beginning of the correction works. At that time, the Rhine was a wide and braiding channel system (width ranging from 500 m to 1,500 m) characterized by numerous in-channel gravel bars, which is consistent with the descriptions from Herget et al. (2005) and Schäfer (1973). Multiple anastomosing channels also existed along the floodplain (Carbiener, 1983), at a maximum distance of about 5 km from the thalweg. The period 1743-1828 is characterized by marked changes and strong channel shifting of about 1 to 2 km. Across the entire study area, gravel bar surface areas increased (+100 ha; +128%) while vegetated areas changed only slightly (low vegetation: -45 ha; -18.4%; high vegetation: +124 ha; +11%; Fig. 4-b). In 1828, more than 70 % of the present natural reserve area was occupied by the active channel (running water and gravel bars). The high morphodynamic activity may be caused by the high frequency of flood





events (four 10-years floods from 1810 to 1828) characterizing the beginning of the 19th century (Fig. 5-c). These dynamics could be an effect of the Little Ice Age (Martin et al., 2015; Schmitt et al., 2016), which may have had a considerable impact on discharge, bedload transport and flood regime intensifying lateral dynamics (Rumsby and Macklin, 1996; Schirmer, 1988). A similar phenomenon has been observed previously, for example, by Bonnefont and Carcaud (1997) on the River Moselle,

or by Bravard (2003) for the Rhône Basin. However, this hypothesis has not been validated for the Rhine yet and awaits further testing (Schmitt et al., 2016; Wetter et al., 2011).

In the scale of the natural reserve, the reach was located on the left bank of the Rhine in 1743. It was almost completely occupied by the thalweg from 1778 to 1828 as it shifted towards the western direction. According to the location of pit 1 (Fig. 4-a, median bar on 1828 map) and the depositional history deduced from historical maps, the resetting of the IRSL signal in

basal sediments of pit 1 resulted from this major lateral migration (Preusser et al., 2016). In accordance with the IRSL dates, deposition of fine grained sediments in the lower part of pit 1 took place between 1778 and 1806 (Fig. 5-a) after the Rhine thalweg had moved over the area. The maximum age of the investigated sediments in pit 1 is therefore 238-210 years (i.e. 1777-1805; Preusser et al., 2016). The 1828 map in figure 4-a shows that pit 2 is located in the 1828 main channel and accumulation of fine sediments must have commenced after this time. Despite local diking on the floodplain and across some

lateral channels as revealed by the planimetric analyses, it seems that fluvial morphodynamics and lateral channel shifts before 1828 were not influenced by anthropogenic disturbances.





**Figure 4: (a)** diachronic evolution of the whole study area. The black-boxes are zooms of (scale 1:2) in the surroundings of the two pits. **(b)** Surface evolutions of the morpho-ecological units in the natural reserve (1743-2010) and in the whole study area (1743-1949). **(c)** Number of anchor points used to georeferenced old maps (1743/1778/1828/1838/1872/1926) and aerial photographs (1949/1956/1966/1971/1985) and values of RMSE errors.





### 4.2 Hydromorphological disturbances during the correction works (1833-1876)

#### 4.2.1 Main channel adjustments

The correction works commenced between 1828 and 1838 (around 1833) and induced major changes of the hydrosystem (Herget et al., 2005). A detailed map based comparison of the 1828 and 1838 maps shows that the left bank was eroded in the

southern part of the study area (maximum of about 50 m) after 1828 and before (possibly during?) the building of the *perpendicular in-channel dike* (Fig. 4-a, 1838). This is in agreement with the current position of the upstream part of the Bauerngrundwasser (Fig. 9, period A). Subsequently, the Rhine thalweg was artificially shifted in a western direction by the *right bank dike,* and then to the east by the *perpendicular in-channel dike* and the *high-flow dike*. A large flood (Q ~ 3800 $m^3.s^{-1}$; 10-years floods; Fig. 5-c) occurred in 1831, which was thoroughly documented due to the important damages it caused

(Champion, 1863 - Testimony of M. Coumes). We hypothesize that this event intensified the morphological adjustments during this period. In 10 years (1828-1838) the surface of running water decreased of 20 ha (-5%) in the entire study area while the surface area of stagnant water increased correspondingly (+75 ha or +85 %).

During the 1838-1872, about 40 years after the beginning of the correction works, the new corrected Rhine channel began to incise (1 cm / year on average; Fig. 5-a) in response to channel narrowing (250 m wide), slope increase and bank stabilization

(Fig. 4-c, 1872). In addition, areas of gravel bars, flowing water and low vegetation surfaces decreased (-110 ha or -70 %; -194 ha or -50 %; -121 ha or -67 %, respectively), while areas of high vegetation and agricultural surfaces increased (+218 ha or +18 %; +183 ha or +700 %, respectively; Fig. 4-b). This is interpreted as the consequence of sediment deposition in the disconnected parts of the main channel, inducing channel narrowing, and forest and agriculture expansion, as intended by the correction works (Bernhardt, 2000). These types of morphodynamic disturbances driven by channel correction were also

observed by David et al. (2016) on the Garonne River, Magdaleno et al. (2012) on the Ebro River and Habersack et al. (2013) on the Danube River.



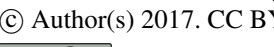

**Figure 5: (a) Vertical evolution of the Rhine thalweg from 1815 to 1960 based on low water levels (Q ~ 540 m³.s⁻¹) recorded at the Marlen gauging station (Bull, 1885; Bensing, 1966; CECR, 1978). Vertical evolution of the pits linked to the age-depth model and number of floods which attained each pits (triangle), (b) location of the two pits and of the ancient gauging station of Marlen, (c) discharge of the Rhine at the gauging station of Basel. The period 1810-1870 corresponds to maximum instantaneous annual flows.**



The period 1870-2015 corresponds to the highest mean daily flow (OFEV: Office Fédéral de l'EnVironnement). The dates with arrows corresponds to old maps or aerial photographs (see figure 4). The red and green lines correspond to the submersion discharges for the pits 1 and 2 respectively, (d) flood return periods at Basel between 1870-2015 (m³.s⁻¹; Adjustment of Pearson III, OFEV).

### 4.2.2    Lateral channel adjustments and filling processes

At the scale of the natural reserve, from 1828 to 1838, surface areas of running water decreased (-21 ha; -13 %), while stagnant water and high vegetation areas increased. In particular, vegetation populated median bars (Fig. 4-a). This general trend became even more marked during the 1838-1872 period (Fig. 4-b), running water and gravel bar areas declined (-65 ha or -47%; -45 ha or -71 %, respectively), and high vegetation and stagnant water areas increased (+90 ha or +101 %; +19 ha or +166 %, respectively). This supports the idea that the sequential impacts induced by the correction works along the floodplain (i.e. channel narrowing, expansion of vegetation) were especially dynamic in the upstream part of the Bauerngrundwasser. The channel filling dynamics are detailed in a *CM* diagram (Fig. 6-a) which shows that graded suspension deposition (QR segment) of sandy loam occurred at the transect 2. In addition, Figure 6-a shows a general and concomitant decrease in grain size and sediment sorting from the bottom to the surface, which reveals a decline in flood energy likely induced by channel diversion following the correction. Furthermore, residual T2 samples are located on RS segment (uniform suspension). They correspond to fine sediments (silt) with sorting and depth are not correlated. These have been deposited on the left bank of the Bauerngrundwasser in low turbulence conditions likely due to site-specific factors such as topography or vegetation, whose general importance has been underlined by Toonen et al. (2015), Bravard et al. (2014) and Riquier et al. (2015). Such kind of depositional filling processes in newly bypassed channels have also been documented by Bravard and Peiry (1999) and Passega (1964, 1977).

The analysis of the evolution of the middle part of the natural reserve from 1838 to 1872, revealed that a large gravel bar was deposited behind the left dike of the corrected Rhine channel (Fig. 4-a). This clearly resulted from an extreme hydrological event, which occurred during the Rhine diking and probably corresponds to the 1852 flood (above 300-years flood; Fig 5-c and Fig. 9, period B). This flood event, referred as the õflood of the centuryö (6.63 m at the Basel gauging station, i.e., 5.78 m higher than the mean low flow water level), has been documented in detail, especially by Pardé (1928), Champion (1863), Eisenmenger (1907), Wittmann (1859). After this flood event, embankment of the Rhine continued but local dike apertures remained open in order to feed some old channels and to enhance filling dynamics (Fig. 4-a, 1872). In addition, the downstream part of the Bauerngrundwasser (middle part of the natural reserve) presented an important connection to the Rhine, until its probable disconnection in 1876 (Casper, 1959; Fischbach, 1878). This specific hydrological condition impacted the depositional filling processes, as shown by the processes observed at the downstream part of the Bauerngrundwasser which are mainly characterized by graded suspension (Fig. 6-a; T4 and T5 transects). Indeed, *CM* diagram shows that energy is higher at T5 than at T4 although T4 is located upstream. Furthermore, the complexity is reinforced by the absence of upward fining in grain size at the surface as shown in the upstream part of the Bauerngrundwasser (Fig. 6-b). Similar to the T2 samples, the position of the other T4 samples in the *CM* pattern corresponds to the mean level of turbulence in the uniform suspension (RS



segment) and are located on the left bank (Fig. 6-a). These T4 samples depend on the great distance from the main channel, as previously observed (Bravard and Peiry, 1999). Generally, the disconnection of the Bauerngrundwasser with the Rhine was reinforced by the total closure of the dike apertures after 1876 (Fig. 4a; 1872, 1926).

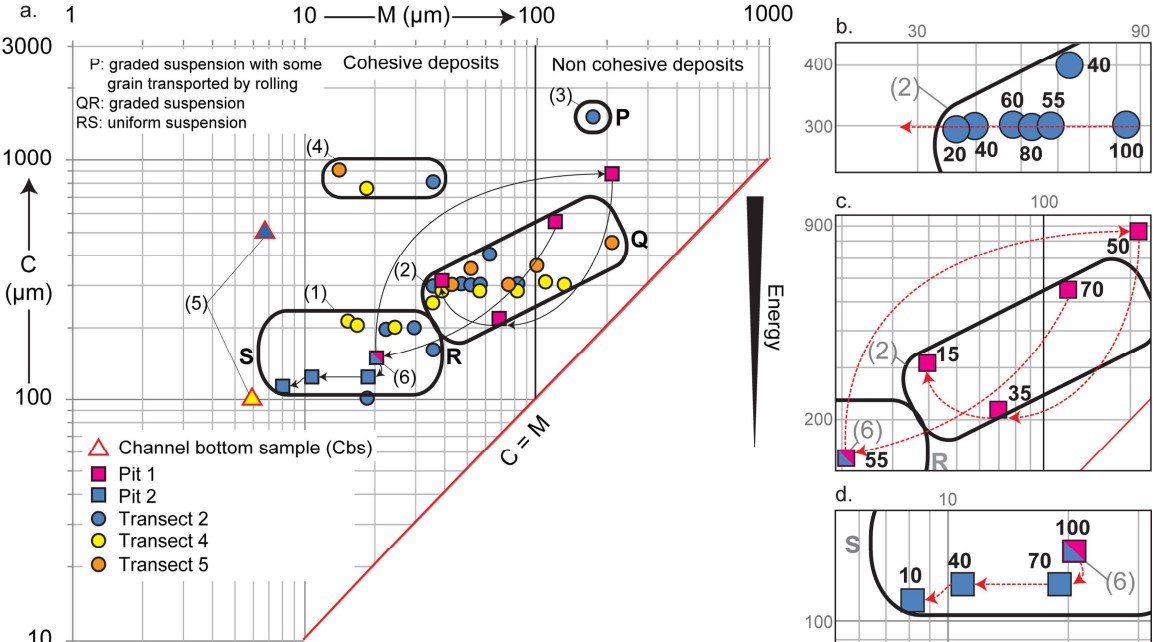

**Figure 6: a. Floodplain *CM* diagram and patterns according to Passega (1964, 1977). Numbers between brackets help for the presentation of the results and the discussion; b. isolated samples with the indication of the samples depth for transect 2, pit 1 (c.) and pit 2 (d.). Red arrows show the deposition chronology.**

On figure 6-a, square tag 6 corresponds of two overlapped samples from the pit 1 – SU 3 and pit 2 – SU 2 (Fig. 7). These samples are composed by the same grain-size characteristics. Square tag 6 is reported on Fig. 6-c & d. Planimetric results

combined with *CM* diagram and IRSL ages show that cohesive sediments which compose the two overlapped samples were deposited during the same flood, probably the 1831 flood. Pit 1 is located on a former large gravel bar in 1828 (Fig. 4-b, 1828). According to our chronology, pit 1 - SU 2 was already deposited at this time (Preusser et al., 2016). This SU is composed of silt deposited during low energy conditions (tag 6; Fig. 6-c). It means that pit 1 likely consists of two depositional filling periods: the first started before 1828 (probably in 1778 for SU 2, 3 & 4; Fig. 7) while the second started with the beginning of

the correction works (for SU 5 & 6, Fig. 7; Preusser et al., 2016). Conversely, pit 2 is located on the 1828 main channel (Fig. 4-b, 1828), which became an overbank area or low energy depositional environments in close vicinity of the main channel in 1838 (Fig. 4-a, 1838). Thus, if any deposits of fine sediments existed in 1828, the filling period in pit 2 have begun after 1828,





as also shown by the IRSL ages (min. : 1828; Fig. 5-a), and was relatively regular. The end of the depositional filling periods at both pits occurred around 1872 as shown by the IRSL ages and is reflected in a grain size refinement and sorting decrease (Fig. 5-a). It coincides with a clear increase in vegetation areas shown by the 1872 map (Fig. 4-a, 1872). Depositional filling differences (periods and processes) between the two pits are mainly controlled by the elevation and the location of the pits in

the floodplain (Fig. 3-c), which in turn determine the frequency of flooding (Fig. 5-a & c) and mean sedimentation rates (from 0.9 cm / year in pit 1 and 6 cm / year in pit 2; Fig. 5-a and Fig. 7). This may also explain the heterogeneity of depositional filling processes at pit 1 (Fig. 6-c) in contrast to pit 2 where grain size generally decreases with increasing elevation; Fig. 6-d). This phenomenon has been observed for disconnected lateral channels by several studies (e.g. Bravard et al.,1986; Hooke, 1995; Riquier et al., 2015).

**4.2.3    Geochemistry of the sediment filling**

Geochronological data combined with geochemical data confirmed changes in the hydrosystem and sediment deposition dynamics of the Rohrschollen from the beginning of the correction works. Quartz ($SiO_2$) was the dominant mineral (range 54 % to 63 %; Fig. 7), and does not show any particular trend down the profile in pit 1 and pit 2. MnO, $TiO_2$ and $P_2O_5$ were least abundant in both pits. This shows that the patterns of mineral composition are not directly related to grain size and likely reflect

a common sediment source to both pits (Grygar et al., 2016). However, lower ratios of Al/Si, especially in the lower sediment layers of SU 2 and 3 in both pits 1 and 2, reflect a general low clay content with dominant sandy (pit 1) or silty (pit 2) sediment textures. This suggests relatively weak soil development and chemical weathering. The uppermost sediment layers of pit 1 and pit 2 differed in organic carbon (C) (16.4 and 27.0 g.kg$^{-1}$, respectively) while organic matter content and organic carbon gradually decrease with depth in both pits. This suggests different temporal trajectories of deposition of organic-rich sediments

on the two sites, i.e. the gravel bar (pit 1) and the Bauerngrundwasser until 1838 (pit 2).

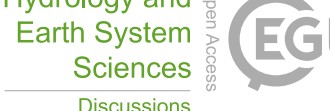



**Figure 7: Stratigraphical log of the two pits including datings, organic contents, sedimentological and geochemical results.**

Regarding pollution histories, polycyclic aromatic hydrocarbons (PAH), polychlorinated biphenyls (PCB), hexachlorocyclohexanes (HCH) and pesticides (Table S1) could not be detected in both pits. This is in agreement with the
5   proposed chronology of pits filling because the first massive use of modern organic pollutants in the whole Rhine catchment gradually increased in the Rhine after 1940, followed by a strong reduction of these pollutants due to the Rhine Action Plan since 1970 (Evers et al., 1988; Goth et al., 2001; Middlekoop, 2000). Similarly, the normalized REE patterns did not show any significant changes with depth in both pits (Fig. S3). Massive use of REE started after 2000 (Klaver et al., 2014), when



sedimentation rates were low on the Rohrschollen compared to the period during and immediately after the correction works (Fig. 5).

To evaluate the potential metal pollution resulting from anthropogenic activities, the relative enrichment in minor element concentrations in the sediments was evaluated in the pits accounting for the geochemical background. Depth-distribution of

minor elements in both pits followed a Sr>Ba>Zr>Cr>REE>Zn pattern. Heavy metal concentrations in pit 1 and pit 2 ranged from 31 to 196 mg.kg$^{-1}$ for Zn, 64 to 91 mg.kg$^{-1}$ for Cr, 10 to 46 mg.kg$^{-1}$ for Cu, 6 to 42 mg.kg$^{-1}$ for Ni, 12 to 35 mg.kg$^{-1}$ for Pb, and 0.23 to 1.24 mg.kg$^{-1}$ for Cd. Cu, Ni, and Cr enrichment factors in pit 1 and pit 2 remained within the natural background (i.e., normalized concentration profiles showed no enrichment in both profiles). This suggests low industrial use of these heavy metals upstream the study site in the 19$^{th}$ century, which is in agreement with previous results (Goth et al., 2001; Middlekoop,

10    2000).

In contrast, Zn enrichment factors decrease with depth in pit 2 (from about 4 to 1), which probably reflects anthropogenic inputs of Zn in the SU 4 to 6 (up to 196 mg.kg$^{-1}$ in SU 6). This is in agreement with previous observations (Ciszewski and Gryar, 2016) insofar metal pollution is expected to be greater at a shorter distance from the main channel thalweg (as for pit 2 until 1838) or from a secondary channel that are hydrologically connected (as for the Bauerngrundwasser after 1838 for pit 2),

compared to a site that is further and less frequently flooded (as for pit 1 after 1828). From about 1860 to the early 1930s, heavy metal concentrations in the Rhine gradually increased in relation to the progressive industrialization (Middelkoop, 2000). Heavy metal enrichment, which is particularly marked for Zn, is therefore likely caused by changes in both heavy metal pollution of the Rhine over the past centuries and sedimentation rates. Although the main depositional filling at pit 2 ended around 1872, it was still flooded at least four times after this date (1876, 1881, 1882 and probably 1910), whereas pit 1 was

more rarely and intensively flooded (1876 and probably 1881; Fig 5-c). Moreover, flow patterns during flooding from 1872 onwards likely controlled sedimentation rates of heavy metal-bound to suspended solids. Overall, results suggest that Zn may be a proxy of anthropogenic deposition histories in the Rhine floodplain, as previously shown in large fluvial systems (Grygar and Popelka, 2016, Lintern et al., 2016).

In addition, water table fluctuation and regular water saturation by flooding may have changed Zn speciation and mobilization

in the SU 1-3 of pit 2, whereas pit 1 remained non-saturated (Fig. 8). In the period between floods, under non-saturated conditions and in the presence of oxygen, oxidation of metal-sulfur, organic carbon and Feó Mn oxyhydroxide in the upper SU may release heavy metals. High flood frequency possibly increased vertical transport of suspended solids and chemical re-distribution in pit 2, which smoothed changes of Zn concentrations with depth (Middelkoop, 1997). During flood events, heavy metals can thus not only move between the SU but also be partly transported into the river water in association with suspended

solids, by local surface erosion (Tao et al., 2005). This emphasizes that, in a restoration context, Zn may be mobilized and transported into the Rhine by groundwater table elevation (Ciszewski and Gryar, 2016) linked to flooding frequency increase as well as by bank erosion.





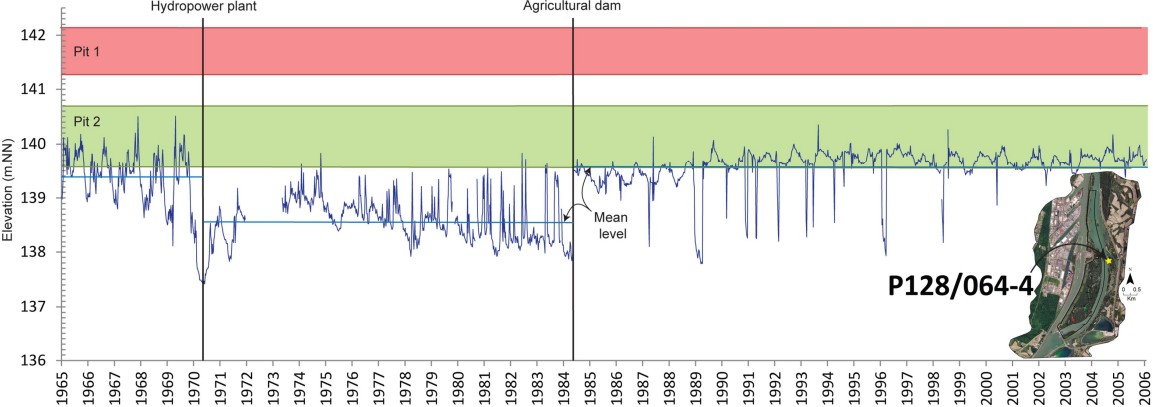

**Figure 8: Hydrogeological variation between 1965 and 2006. After 1984, the lowest peaks correspond to periods during which the Old Rhine is empty.**

### 4.3 Adjustments since the end of the correction works (after 1876)

Results of the *CM* pattern (Fig. 6-a) shows one sample in group 3 and three samples in group 4 that correspond to the left bank of the Bauerngrundwasser (surface layer). This last depositional filling period is linked to several large floods (1876, 1882; > 300 and 100 years return periods, respectively; Fig. 5-c) of high transport energy leading to suspension (Fig. 6-a, group 4) and rolling processes (Fig. 6-a, group 3-P). The 1876 flood, which was the major hydrological event during this period, fed the Bauerngrundwasser from upstream. The flood induced relative high flow conditions with increased sediment transport

capacities, which caused grain size decrease along the channel. This process was particularly pronounced in the coarser P sample, which was deposited on the left-bank of the channel close to T2 (Fig. 4-a, 1926). As identified on the 1926 map, the flood probably opened a dike near the upstream part of the Bauerngrundwasser (Fig. 4-a), which increased flow through the channel. During the same period (1872-1926), last major planimetric and geomorphic changes occurred downstream the Bauerngrundwasser close to the transect 6 (Fischbach, 1878 - Testimony of M. Schanté). In this area, the 1876 flood breached

the downstream part of the *high-flow dike* (Fig. 4-a, 1872, 1926), and widened and accentuated channel bends (Fig. 9, period C). The present morphology of the Bauerngrundwasser results from this event. After this flood, all dike apertures were closed, thereby reinforcing the disconnection between the floodplain and the main channel. This situation was exacerbated by the progressive incision of the corrected Rhine channel (Casper, 1959). Gravel bar surfaces were progressively covered by high vegetation (more than 60 % of the natural reserve area in 1926; Fig. 4-b).

The regularization works (1930-1936) induced a last/second phase of incision (Casper, 1959; Marchal and Delmas, 1959), accentuating the gradual conversion of the Bauerngrundwasser into a wetland (+ 7 ha). At the same time gravel bars appeared on the groyne fields which extended into the corrected Rhine channel (+ 13 ha between 1926 and 1949; Fig. 4-b). In 1970, this Rhine channel was by-passed by the Rhine canal on which a power plant was constructed. A continuous discharge up to 1,550 $m^3.s^{-1}$ is diverted towards the Rhine canal, altering the hydrology of the Old Rhine River drastically (Fig. 5-c, blue hydrogram).





Therefore, groundwater level was lowered by about 0.8 m (Fig. 8). In 1984, the construction of the agricultural dam raised and stabilized the water level of the Old Rhine River at 140.00 m.NN, and the groundwater level around 139.60-140.00 m.NN, while the amplitude of groundwater fluctuation decreased suddenly from 1.5 to 0.4 m; Fig. S3). The entire Bauerngrundwasser was impacted by this backwater effect, which increased the stagnant water surface (+7 ha in 2010; Fig. 4-b). To avoid drying

5 of the Bauerngrundwasser during low flow and drought periods along the Old Rhine River, an input of 1.5 m$^3$.s$^{-1}$ from the Rhine canal feeds the channel by a siphon (Fig. 4-b, 2010, siphon). This explains the specific modern dynamics of sediment suspension at the bottom of the Bauerngrundwasser (group 5; Fig. 6-a). Energy decreases along the channel as shown by the differences between T2-Cbs (C = 500; M = 7) and T4-Cbs (C = 100; M = 6). Similar dynamics were also observed by Peiry (1988) and underscore that in-channel deposition and filling was a current process until the start of the restoration.

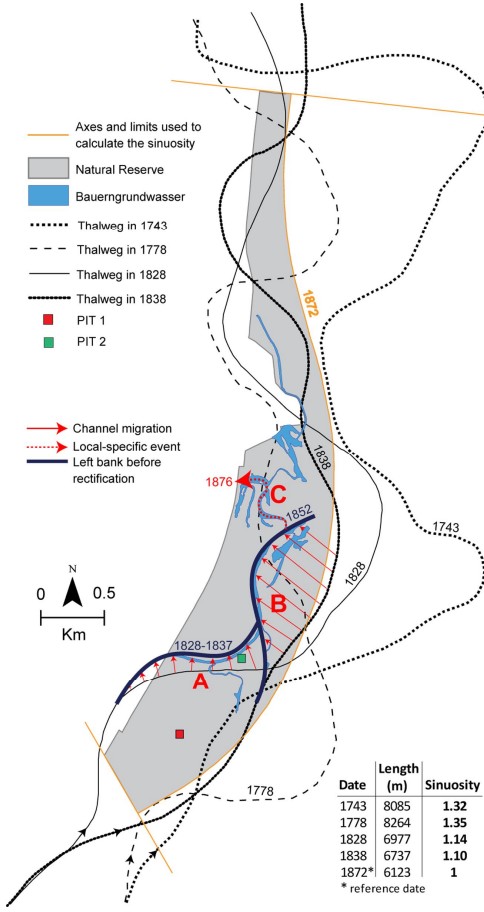

**Figure 9: Rhine thalweg evolution from 1743 to the end of the correction works in 1876. (A) Area of lateral channel mobility during the 1828-1837 period. (B) Area of lateral channel mobility during the 1851-52 flood. (C) Area impacted by the 1876 flood.**



### 4.4 Using Rhine long-term trajectory to enhance efficiency and sustainability: learning from the past to infer restoration guidelines

Combining spatial and temporal scales by overlapping multiple data sources in an interdisciplinary approach appears of crucial importance to (i) determine pre-disturbance dynamics of the hydrosystem, (ii) improve the understanding of the history of the

hydrosystem in fine spatio-temporal scales (Belletti et al., 2014; Bouleau and Pont, 2014; Brierley and Fryirs, 2008), and (iii) provide key information to manage restoration projects in efficient and sustainable ways. Indeed, these results allow validating our three working hypotheses:

> *1. Long-term temporal trajectories allow to identify driving factors, amplitude and response-time of disturbances*

The pre-disturbance functioning of the Rhine hydrosystem in our study area, as identified by historical map analysis,

sedimentological and geochronological data, was characterized by a high energy depositional environment with active braiding, lateral channel mobility and important surface areas of gravel bars and pioneer vegetation. In part, this functioning has been targeted by recent restoration efforts. Our results also highlight that the system has been drastically disturbed from the beginning of the correction works. In addition, the flood regime is an additional and crucial driving factor, inducing important and rapid changes in morphodynamics and sediment deposition, especially in low elevation areas where floods are

relatively frequent, long and intense (palaeochannels; e.g. pit 2). The hydrosystem remained morphologically sensitive to floods throughout the duration of the correction works. From the end of the correction, changes were less marked and also related to a decrease in flood energy and frequency. The volume of fine sediment deposition is higher in the upstream part of the Bauerngrundwasser, with a maximum thickness of 1.8 m and a volume of about 400,000 m$^3$ (estimated by combining diachronic planimetric analysis with thickness of sediment layer close the Bauerngrundwasser). A volume of about 800,000

m$^3$ of fine sediments with a mean thickness of about 0.4 m was deposited in the entire Rohrschollen Island up to today. This highlight the fact that impacts of correction work and further engineering works are irreversible. Hence, the main challenge of the restoration was to recover processes as flood dynamics and a morphodynamic gravel bed channel in a restricted environment.

> *2. Assessing potential benefits and limits of the restoration*

Fine sediments are mainly located along the Bauerngrundwasser and represent a limitation for restoration purposes, because of the risk of their remobilization from both banks and channel bottom. As shown by Boulton et al. (1998) and Richards and Bacon (1994), fine sediments (e.g. sand) are a limiting factor for many aquatic biological species. This point is important because fine sediments were relatively scarce in the braiding-anastomosing Rhine hydrosystem previous to the initiation of correction works (Ochsenbein, 1966). Furthermore, another risk concerns potential remobilization of pollutants bound to fine

sediments. This risk has also been identified and characterized on the Rhône River by Bravard and Gaydou (2015), Desmet et al. (2012) and Provansal et al. (2012). However, partial backwater effects induced by the agricultural dam control the water level on the Bauerngrundwasser, even during ecological floods (partly), and thus may limit bank erosion (Eschbach et al. submitted). Thus, in the specific case of the Rohrschollen Island, both risks are lowered by this local hydraulic constraint.



Conversely, the new channel dug on a large former gravel bar (Fig. 1-d) exposes a thinner layer of fine sediments along its banks, which are mainly composed of coarse sediments (gravels, pebbles) inherited from the pre-correction Rhine and represent a historical in-channel gravel bar. The backwater effect of the agricultural dam does not affect this channel (except the 100 downstream meters; Eschbach et al., 2017). Consequently, the restoration of this channel induces a recovery of bedload

dynamics, lateral channel mobility, morphodynamic and habitat diversification, which stimulate notably key-processes as downwelling/upwelling hydrological exchanges (Eschbach et al., 2017).

>    3.  *Provide key-information to manage functional restoration actions in order to maximize efficiency and sustainability,*
>        *and infer future evolutionary trends*

Efficiency and sustainability are key issues in restoration projects (Bouleau and Pont, 2014; Loomis et al., 2000). For

management strategies, knowledge of temporal trajectories is relevant for targeting pre-disturbance processes (Cairns, 1991) and performing hydromorphological process-based restorations (Mika et al., 2010; Rinaldi et al., 2015). It also allows to identify floodplain areas with high hydromorphological functional potentials, i.e. sectors with thin layers of fine sediments located outside palaeochannels, notably on former gravel bars. In such geomorphological areas, where the efficiency of restored lateral channels may be the highest, managers are encouraged to excavate new channels and enhance morphodynamics

by floods, which even may erode self-formed lateral channels in some cases, rather than reconnecting filled palaeochannels directly. However, the latter restoration measure has dominantly been carried out on large rivers including the Rhine, which raises the question regarding its wider relevance (Schmitt et al., 2009; Schmitt et al., 2012).

Concerning the floodplain compartment, the sedimentation rate is relatively low on the Rohrschollen Island (about 0.1 cm/y), but it is higher (of about one order of magnitude) on areas where flood intensities and frequencies have been less impacted

than the Rohrschollen Island, along the non-canalized section of the Upper Rhine (Frings et al., 2014; Dister et al., 1990) and in the Rhine delta (Hudson, 2008). This floodplain geomorphological evolution reduces flood retention capacities. It probably will require in the future innovative flood management strategies (Hudson, 2008) that may notably be based on floodplain artificial excavations of fine sediments. This also should allow recovery of coarse sediments in the floodplains, a texture which is currently lacking. The long term (> 100 years) sustainability of the hydro-geomorphological management of the Upper

Rhine appears as a key-question, which is coming more and more important, and is complexified by possible fine sediment pollutions.

Morphodynamic (and ecological) fluvial functionality requires relatively high, intense and long flood events superimposed on the natural hydrological regime (Bayley, 1991; Dister, 1992). This is the case on Rohrschollen Island, but following the first flooding events after restoration, a further question arises, among other: how to manage such a highly dynamic environment

on median/long terms? For example, it will probably be necessary to equilibrate the Lane balance of the new channel by upstream artificial gravel augmentations, in the next years/decades. The sustainability of functional restoration efforts and their management remains an open question: on which time scale, for which compartment and over which spatial scales should they be considered? In this context, it appears crucial to continue the post-restoration monitoring over medium (3-5 years) and long (> 5-10 years) time scales to evaluate restoration success (Jähnig et al., 2011; Kondolf and Micheli, 1995; Palmer et al., 2005).



This opens up avenues for developing integrative methodological approaches to improve pre-restoration knowledge and to implement post-restoration monitoring.

## 5    Acknowledgments

This study has been funded by the European Community (LIFE08 NAT/F/00471), the City of Strasbourg, the University of
Strasbourg (IDEX-CNRS 2014 MODELROH project), the French National Center for Scientific Research (CNRS), the ZAEU (Zone Atelier Environnementale Urbaine - LTER), the Water Rhine-Meuse Agency, the DREAL Grand Est, the õRégion Alsaceö, the õDépartement du Bas-Rhinö and the company õÉlectricité de Franceö. We acknowledge Arthur Zimmermann for the GIS implementation, Jérôme Houssier and Erni Dillmann for handing over the historical maps, Martine Trautmann for the grain size analyses (EOST-UMS 830), Pascal Finaud-Guyot for reviewing the flooding frequency assessment, Claire Rambeau
for field assistance, and Ferréol Salomon and Christian Damm for fruitful scientific discussions.

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
