# Peer review of "Long-term river trajectories to enhance restoration efficiency and sustainability on large rivers: an interdisciplinary study"

_Hydrology and Earth System Sciences, 2017_

## Referee Comment (RC1) · Anonymous Referee #1 · 11 Oct 2017

GENERAL COMMENTS The study deals with the long-term evolution of a reach of the Rhine River which underwent some restoration activities. Overall, I think it is a very good work: the novelty is combining reconstruction of morphological evolution with other aspects, specifically with geochemical characteristics of sediments. Some revisions are needed to make some parts more effective, especially the last section (see "specific comments"), and to put the work in a wider context (it could be useful to summarize one or two key points that comes out from this study and this restoration project).

[Figure]

SPECIFIC COMMENTS "Study area" section (pages 4 and 5). The part dealing with the restoration project could be improved. I think it could be useful to describe a little bit more in detail the restoration project and, in particular, the aims of the project. This would be very helpful for improving the last section of the manuscript (4.4) (see one of the following comments).

Page 8, L. 2. A brief explanation of the CM diagram method would be useful.

IRSL dating. I have some concerns about using this dating method within this study: is this method appropriate to the temporal scale considered in this study? How much reliable are the results? I am specifically referring to Figure 7, which shows that dates have significant errors and reverse ages can be obtained (see pit 2, where there is a reverse relation between sediment depth and age). Overall the contribution of IRSL may be considered useful for this study, since it constrains the age of fine sediment deposition, but it would be useful if authors would add some comments on such data. For instance, could alternative dating method be used in a similar context?

Section 4.4. This is part could be improved: considering the amount and quality of data, I think that the authors could make some efforts to make this part more effective. I think that they should try to go more in detail about the effects of the restoration project. For instance: were the project aims appropriate for this river reach? To what extent are (or will be) those aims achieved? Which are the main limitations of a restoration project carried out at reach scale, such as the one described in this study? Other examples to improve this section: "in part, this functioning has been targeted by recent restoration efforts" (Page 21, L. 11-12), this could be illustrated more in detail; "this highlight the impact...works are irreversible" (Page 21, L. 20-21), this statement requires further explanations. I am wondering if it could be useful to add a final section (e.g. "Conclusions" or "Final remarks") where major outcomes of this study (both specific and general) could be summarized.

Page 22, L. 30. This sentence is not clear: I think it would be useful to explain better what could be likely the future evolution of this reach, and I would avoid a direct reference to Lane balance (it is a concept well known among geomorphologists but, probably, not for readers with different backgrounds).

Figure 9. I have some concerns about this figure. Is it really meaningful to calculate sinuosity if channel configuration was multi-thread from 1743 to 1838? Sinuosity is a key characteristic in single-thread channel, while less relevant in multi-thread channel. I think that it is not correct to assume that sinuosity in 1872 was 1 (it does not look like a straight channel!). I am wondering if this figure could be removed.

Some suggests concerning terminology: "channelization" or "channelization work" instead of "correction"; Page 14, L. 7. "Central bar" instead of "median bar".

TECHNICAL CORRECTIONS Page 1 – L. 21. "IRSL" instead of "IRLS" Page 2 – L. 1. It could be better to use a chronological order where several works are cited. Please consider this comment throughout the manuscript. Page 4 – L. 15. Figure 3 (as well as Figure 4, page 5 L. 7) is cited within the main text before Figure 2. Page 20, L. 2. What is the meaning of "NN"? Above sea level? Page 21, L. 5. "Different spatio-temporal scales"? Page 21, L. 32. Eschbach et al., submitted is missing in the reference list. Page 22, L. 33. "Short" instead of "medium"? Figure 4c. A legend should be add to explain the two symbols of this figure (i.e. anchor points and RMSE errors).

---

## Author Comment (AC1) · 17 Jan 2018

-> Please note that the title will be modified by "Long-term river trajectories to enhance restoration efficiency and sustainability: an interdisciplinary study (Upper Rhine, France)"

GENERAL COMMENTS

The study deals with the long-term evolution of a reach of the Rhine River which underwent some restoration activities. Overall, I think it is a very good work: the novelty is combining reconstruction of morphological evolution with other aspects, specifically with geochemical characteristics of sediments. Some revisions are needed to make some parts more effective, especially the last section (see "specific comments"), and to put the work in a wider context (it could be useful to summarize one or two key points that comes out from this study and this restoration project). -> For this last part outlined in bracket, we will summarize the main findings of the study in a short conclusion.

SPECIFIC COMMENTS

"Study area" section (pages 4 and 5). The part dealing with the restoration project could be improved. I think it could be useful to describe a little bit more in detail the restoration project and, in particular, the aims of the project. This would be very helpful for improving the last section of the manuscript (4.4) (see one of the following comments). -> This will be done accordingly.

Page 8, L. 2. A brief explanation of the CM diagram method would be useful. -> This will be done accordingly.

IRSL dating. I have some concerns about using this dating method within this study: is this method appropriate to the temporal scale considered in this study? -> Indeed, the presented IRSL ages are close to the upper dating range of the method, but it has been shown in previous papers that ages of a few years can reliably be determined using luminescence methods (e.g. Ballarini et al. 2003, Quat. Geochron.; Madsen et al. 2005, Marine Geol.).

How much reliable are the results? -> This question applies in general to geochronological data, actually to any kind of data collected. For the present samples, we refer to the detailed discussion in Preusser et al. (2016, Geochronometria).

I am specifically referring to Figure 7, which shows that dates have significant errors

and reverse ages can be obtained (see pit 2, where there is a reverse relation between sediment depth and age). -> We cannot follow the reviewer here. What are 'significant errors'? The ages are associated with uncertainties of ca. 10 %, i.e. lower than those related to radiocarbon dating in this time range due to calibration uncertainties. As a matter of fact, there are no reverse ages in the study, see figure 7. The ages are all in excellent agreement within uncertainties.

Overall the contribution of IRSL may be considered useful for this study, since it constrains the age of fine sediment deposition, but it would be useful if authors would add some comments on such data. For instance, could alternative dating method be used in a similar context? -> Again, we refer to the paper by Preusser et al. (2016), providing a full overview of the topic and discussing the issues raised by the reviewer. Since such a discussion is well beyond the scope of the present study, we consider inappropriate to repeat it here. In summary, there is no alternative dating approach.

Section 4.4. This is part could be improved: considering the amount and quality of data, I think that the authors could make some efforts to make this part more effective. I think that they should try to go more in detail about the effects of the restoration project. For instance: were the project aims appropriate for this river reach? -> The effects of the restoration project cannot be developed here. The aim of this section is to highlight how long-term trajectory of the hydrosystem allows identifying the driving factors, amplitude and response-time of past disturbances. We show that this study is unavoidable and contributes to the construction of a restoration project. The effects of the restoration linked with the inherited morphological characteristics are developed in another paper submitted in Geomorphology. However, we will improve the text to highlight the legitimacy (efficiency and sustainability) of the restoration project, which is strengthened by considering the historical context.

To what extent are (or will be) those aims achieved? -> Again, this is developed in the paper submitted in Geomorphology. A retrospective analysis is carried out to determine the efficiency of the restoration project in the basis of a fine monitoring which is leading

in short term (3 years).

Which are the main limitations of a restoration project carried out at reach scale, such as the one described in this study? -> Main limitations are developed in section 4.4.2. For instance, we showed that fine sediments could be remobilized as well as pollutants bound in such sediments.

Other examples to improve this section: "in part, this functioning has been targeted by recent restoration efforts" (Page 21, L. 11-12), this could be illustrated more in detail; -> Of course, by "functioning" we mean "the functioning before major engineering works", and by "restoration efforts", we mean the creation of hydromorphological dynamics. Aims of the restoration are twofold: reinstalling lateral and vertical dynamics into the channel and stimulating bedload dynamics and groundwater - surface water exchanges. We propose to add the following sentence: Âń Indeed, this functioning and processes have been targeted by recent restoration efforts": "More specifically, the restoration aims have been to recover bedload transport, lateral and vertical dynamics, as well as groundwater - surface water exchanges".

"this highlight the impact: "works are irreversible" (Page 21, L. 20-21), this statement requires further explanations. -> We propose to give more information by adding the following text after "irreversible": "... because the removal of very large amounts of fine sediments seems unthinkable. Furthermore, the strong hydrological alteration by the canalization works makes the functional alteration of the hydrosystem irreversible as well. In this context, the main challenge...". Following this idea, we will add the following text p. 22, L.23, after "...fine sediments": "... and/or natural fine sediment removal by the recovery of active bank erosion in lateral channels".

I am wondering if it could be useful to add a final section (e.g. "Conclusions" or "Final remarks") where major outcomes of this study (both specific and general) could be summarized. -> This will be done.

Page 22, L. 30. This sentence is not clear: I think it would be useful to explain better what could be likely the future evolution of this reach, and I would avoid a direct reference to Lane balance (it is a concept well known among geomorphologists but, probably, not for readers with different backgrounds). -> This will be done. We will add the following text: ". . .years/decades, in order to consider a likely future sediment deficit in the upstream extremity of the channel".

Figure 9. I have some concerns about this figure. Is it really meaningful to calculate sinuosity if channel configuration was multi-thread from 1743 to 1838? -> In braiding systems, sinuosity can be calculated by using the thalweg of the main channel.

Sinuosity is a key characteristic in single-thread channel, while less relevant in multi-thread channel. I think that it is not correct to assume that sinuosity in 1872 was 1 (it does not look like a straight channel!). I am wondering if this figure could be removed. -> The average axis of the active band (which corresponds roughly here to the 1872 channel) can be used as a reference to calculate the sinuosity, as this was also shown by Malavoi and Bravard (2010). Moreover, this figure cannot be removed because it summarizes the morphological planimetric evolutions driven by the engineering works, including the sinuosity.

Some suggests concerning terminology: "channelization" or "channelization work" instead of "correction"; -> "Correction" or "correction works" are the specific terminology used in engineering reports, books or articles to consider the first engineering works in the Upper Rhine River (project of Tulla). That explains why we preferred this terminology as "channelization". Moreover, this is previously specified in the introduction.

Page 14, L. 7. "Central bar" instead of "median bar". -> This will be done.

TECHNICAL CORRECTIONS

Page 1 – L. 21. "IRSL" instead of "IRLS" -> This will be done.

Page 2 – L. 1. It could be better to use a chronological order where several works are cited. Please consider this comment throughout the manuscript. -> This will be done.

Page 4 – L. 15. Figure 3 (as well as Figure 4, page 5 L. 7) is cited within the main text before Figure 2. -> We will remove the references to these figures.

Page 20, L. 2. What is the meaning of "NN"? Above sea level? -> NormalNull is a specific altimetric system reference used in the Upper Rhine. This means that the sea level reference is located in Germany (North Sea) and not in France (Mediterranean Sea).

Page 21, L. 5. "Different spatio-temporal scales"? -> We propose to keep this expression because it is commonly used to describe an interlocking of spatial scales and temporal scales

Page 21, L. 32. Eschbach et al., submitted is missing in the reference list. -> This will be done.

Page 22, L. 33. "Short" instead of "medium"? -> This will be done.

Figure 4c. A legend should be add to explain the two symbols of this figure (i.e. anchor points and RMSE errors). -> In the figure 4c, the legend of the bold and dotted lines refer to the line style used for the vertical axis.

---

## Referee Comment (RC2) · Anonymous Referee #1 · 10 Feb 2018

I am disappointed by some of the author's replies, also considering that my comments were not too difficult to address and aiming to improve quality and clarity of the paper. Specifically, I think that the author could pay more attention to the following aspects: (1) Dating by IRSL. The authors replied that there are not reverse ages: is PIT 2 showing a "normal" relation between depth and ages of sediments (figure 7)? I understand that this could be the only option for dating: on the other hand, I think it could be useful to say that there were no other options. (2) Section 4.4. The authors replied that "...effects of the restoration project cannot be developed here...". If so, why in the

"Introduction " they say "…and to assess potential benefits and limits of the restoration" (page 3, line 5) and "…evaluate efficiency and sustainability of the restoration effects…" (page 3, line 13)? I understand that they want to avoid overlapping between this paper and another one submitted to "Geomorphology": in this case, my suggest would be to make some change in the "Introduction" to make the whole work more consistent. (3) Sinuosity (Figure 9). Yes, I agree that sinuosity can be measured in a braided rivers: the point is that if you are analyzing a multithread river (braided, wandering; see figure 4) other indices would be more useful to be taken into account (e.g. braiding index).

---

## Editor Comment (EC1) · HHG Savenije (Editor) · 12 Feb 2018

Dear Authors,

I am sorry that the review of your paper took so long. It was hard to find reviewers, also because your paper addresses river engineering issues rather than hydrological. But the main reason why it took so long is because you took a long time to react to the first review by Referee#1. Only after several reminders from my side, you wrote a reply, which I found perfunctory and disappointing. The Referee#1 also indicated this in the

reply on your response.

This is not to say that your work is not interesting for the journal. That is why I'll decide to accept your paper only after modifications are made that fully satisfy the observations made by Referee#1. Please take time to do this to our full satisfaction. After you have submitted your revised version, I shall ask the Referee#1 again if he/she is satisfied with your handling of the comments. Depending on this feedback I shall make a decision on acceptance or rejection.

Sincerely,

Hubert Savenije

editor
* * *

---

## Author Comment (AC2) · 8 Mar 2018

Dear Hubert Savenije,

First, we profoundly thank you for your help to improve our paper and give us the opportunity to make modifications on it. We are sorry to have taken a long time to react to the first review by Referee#1. Indeed, the last months were very busy because I finished my PhD at the same time and I have taken up a new position.

We have carefully considered and implemented the helpful suggestions made by the

Referee#1 in RC1 and RC2. We would like to thank you for having given us the opportunity to submit a revised manuscript and look forward to the next steps in the evaluation process of our contribution. We remain at your disposal for any further information that would be required.

Yours sincerely,

David Eschbach Corresponding author, on behalf of all co-authors
* * *

---

## Author Comment (AC3) · 8 Mar 2018

R1: I am disappointed by some of the author's replies, also considering that my comments were not too difficult to address and aiming to improve quality and clarity of the paper.

AR: We thank you sincerely for your helpful remarks aiming to improve the paper, in quality and clarity. We are profoundly sorry that you considered our first corrections not satisfactory. We have carefully considered all your comments (RC1 & RC2) and

have done our best to satisfy your and the editors requests. We sincerely hope our corrections and the revised version will be considered positively.
* * *
R1: Specifically, I think that the author could pay more attention to the following aspects:

(1) Dating by IRSL. The authors replied that there are not reverse ages: is PIT 2 showing a "normal" relation between depth and ages of sediments (figure 7)? I understand that this could be the only option for dating: on the other hand, I think it could be useful to say that there were no other options.

AR: The IRSL ages shown in Figure 7 for PIT 2 are $179 \pm 35$ (at 40 cm), $170 \pm 26$ (58 cm) and $165 \pm 22$ (97 cm). First, it is important to note that these ages all overlap well within the given uncertainties. For a set of three ages there are n * (n-1) possibilities (6 in our cause) of how these ages can be arranged. The end members would be 1-2-3 and 3-2-1 (where 3 is the youngest and 1 is the oldest age). We have computed the likelihood for the different possible combinations based on their uncertainties using the following MATLAB code (by courtesy of Prof. S. Hergarten, Freiburg):

```
n = 1000000; a = randn(3,n); a(1,:) = 179 + a(1,:) * 35; a(2,:) = 170 + a(2,:) * 25; a(3,:) = 165 + a(3,:) * 22; [~,index] = sort(a); index = 100*index(1,:)+10*index(2,:)+index(3,:); h = hist(index,1:1000); % probabilites of the 6 possible orders in percent; h = 100*h(h~=0)/n;
```

For this we received the following results:

% 1 2 3 / 1 3 2 / 2 1 3 / 2 3 1 / 3 1 2 / 3 2 1

% 11.47 / 15.22 / 10.08 / 22.51 / 15.00 / 25.72

This indicates that the observed order of values has a likelihood of more than 10% and, indeed, we are considering this a statistically likely enough case to be regarded

as "normal" (from a statistical point of view). In other words, the only information we can deduct from the dates of PIT 2 is that they reveal a rapid phase of sediment deposition - with an absence of a significant correlation between depth and ages -, which corresponds to the phase of correction works of the Rhine on the Rohrschollen site.

AC: Furthermore, in order to follow your suggestion, we propose to add a sentence to specify that the IRSL method is the only option for dating in this context. Section 3.3.3: ", . . . as any other alternative approach is achievable in this context (Preusser et al., 2016)", after "Dating of sediments sampled from both pits was carried out using Infrared Stimulated Luminescence (IRSL). . .".
* * *
R1: (2) Section 4.4. The authors replied that ". . . effects of the restoration project cannot be developed here. . .". If so, why in the "Introduction " they say ". . .and to assess potential benefits and limits of the restoration" (page 3, line 5) and ". . .evaluate efficiency and sustainability of the restoration effects. . ." (page 3, line 13)? I understand that they want to avoid overlapping between this paper and another one submitted to "Geomorphology": in this case, my suggest would be to make some change in the "Introduction" to make the whole work more consistent.

AR: We fully understand the remark of Reviewer#1 and we thank him for this helpful comment. From a general point of view, the aim of the paper is to show that long-term trajectory of the hydrosystem is useful to improve the efficiency and the sustainability of the Rohrschollen restoration project (and potentially of other river restoration projects). This is presented in the introduction, as well as in the discussion (section 4.4). We discuss the general knowledge produced thanks to the historical study, and which is useful in the restoration context (limits, benefits, efficiency, sustainability), but we don't aim to show the post-restoration changes observed in the Rohrschollen Island with large detail, because we believe that (i) this would be beyond the scope and topic of the paper, and (ii) would make the paper unnecessarily long. Furthermore, these detailed results

have been published by Eschbach et al. (2017) and may be published in the near future (Eschbach et al., in review). Nevertheless, we fully agree with Reviewer#1 (RC1 and RC2) that the restoration project and the morphological evolution of the restored channel should be better presented in the Study area section, notably to strengthen the discussion. Consequently, we propose to add / modify several sentences in the introduction and the discussion, both modifications being linked together.

AC: End of the Introduction, we propose to modify the objectives in order to make them coherent with the discussion ("evolutionary trends" is more general than "evolution"; the objective (v) opens avenues towards other river restorations): "..., (iv) deduce post-restoration evolutionary trends and (v) propose operational outlook to improve efficiency and sustainability of Rohrschollen's restoration, and by extension of other river restoration projects (Sear et al., 1994; Grabowski and Gurnell, 2016)", after "...(iii) characterize assess physio-chemical sediment properties (e.g. heavy metals and organic contaminant concentrations) to trace anthropogenic activities and evaluate the potential impact of the restoration on pollutant remobilization (Middelkoop, 2000; Fedorenkova et al., 2013; IKSR-CIPR-ICBR, 2014; Middelkoop, 2000).". At the end of the Study area section, we propose to add some details on the restoration and post-restoration adjustments: "As the bankfull discharge of the new channel is 20 m3.s-1, flooding in the Island occurs when the discharge exceed this threshold. A three years monitoring showed that bedload transport, active lateral and vertical morphodynamics occur along the new channel (active bank erosion, formation of bars and logjams, enhancement of pool-riffle sequences, increase of groundwater – surface water exchanges...; Eschbach et al., 2017; Eschbach et al., in review), but not along the Bauerngrundwasser which is affected by the hydraulic backwater of the agricultural dam (Eschbach et al., 2017; Eschbach et al., in review; see also the pictures of Fig. 1).", after "Water input from the flood gate ranges between 2 m3.s-1 (when Q Rhine < 1550 m3.s-1) and 80 m3.s-1 (when Q Rhine > 1550 m3.s-1).".

In the Discussion, Section 4.4. §1, we propose to add: "More specifically, the restoration induced, in the new channel, the recovery of bedload transport, lateral and vertical dynamics, as well as groundwater - surface water exchanges.", after "In part, this functioning has been targeted by recent restoration efforts.".

Section 4.4. §1, we propose to complete (two more sentences) and modify the end of the section: "... because the removal of very large amounts of fine sediments seems unthinkable. Furthermore, the strong hydrological alteration by the canalization works makes the functional alteration of the hydrosystem irreversible as well. In this constrained context, the main challenge of the restoration was to recover processes as dynamic floods (on the whole island) and a morphodynamic gravel bed channel in a relatively restricted environment (new channel; see also below). On the basis of an environmental monitoring conducted during three years after the end of the restoration works, it appears that these restoration objectives are attained (Eschbach et al., 2017; Eschbach et al., in review) and that the restoration choices were relevant (see also below).", after "This highlight the fact that impacts of correction work and further engineering works are irreversible...".

Section 4.4. §2, we propose to add: "This demonstrates once again the relevancy of the principles of this restoration.", after "Thus, in the specific case of the Rohrschollen Island, both risks are drastically lowered by this local hydraulic constraint.".

Section 4.4. §3, we propose to add: "... as it is the case on the Rohrschollen Island (new channel).", after "It also allows to identify floodplain areas with high hydromorphological functional potentials, i.e. sectors with thin layers of fine sediments located outside palaeochannels, notably on former gravel bars, ...".

Section 4.4. §3, we propose to add: "... (as it has been performed on the Rohrschollen Island),...", after "In such geomorphological areas, where the efficiency of restored lateral channels may be the highest, managers are encouraged to excavate new channels and enhance morphodynamics by floods, which even may erode self-formed lateral channels in some cases...".

Section 4.4. §3, we propose to add: ". . . (which may be impossible if sediments are polluted) and/or natural fine sediments removal by the restoration of active bank erosion in lateral channels.", after "It probably will require in the future innovative flood management strategies (Hudson, 2008) that may notably be based on floodplain artificial excavations of fine sediments. . .".

Section 4.4. §3, we propose to add: ". . . to balance a relative sediment deficit in the upstream section of the new channel by artificial gravel augmentations, in the next years/decades (Eschbach et al., in review).", after "For example, it will probably be necessary. . .".

Section 4.4. §3, we propose to add: ". . .and modelling, both in the frame of fluvial hydrosystem temporal trajectories.", after "This opens up avenues for developing integrative methodological approaches to improve pre-restoration knowledge and to implement post-restoration monitoring. . .".
* * *
R1: (3) Sinuosity (Figure 9). Yes, I agree that sinuosity can be measured in a braided rivers: the point is that if you are analyzing a multithread river (braided, wandering; see figure 4) other indices would be more useful to be taken into account (e.g. braiding index).

AR: In order to take into account the remarks RC1 and RC2 of the Reviewer#1, we have calculated the thalweg's sinuosity on the basis of the straight length of the reach (Fig. 8). So, the sinuosity in 1872 is 1.09 rather than 1.00, which was effectively wrong. Furthermore, we have added a Braiding and Anastomosing Index (BAI; table of Fig. 8) which corresponds to the mean number of these two types of channels, (channels showing stagnant water have been excluded). Indeed, this index shows in a relevant way the modifications of the channel pattern (BAI decreased from 7.90 to 1.00).

AC: As a consequence, the following changes in the text are also proposed:

Section Study area, we propose to add: "Before engineering works, it was a braiding and anastomosing fluvial hydrosystem.", after "The Rohrschollen artificial Island is located 8 km South-East of the city of Strasbourg and owes its existence to the construction of a power plant in 1970.".

Section 4.1., we propose to add: "The braiding and anastomosing index ranged between 7.9 and 5.4 (Fig. 8)", after "...5 km from the thalweg.".

Beginning of the section 4.2.2., we propose to add: "..., the braiding and anastomosing index decreased from 5.36 to 2.45", after "At the scale of the natural reserve, from 1828 to 1838...".

Section 4.2.2., we propose to add: "(...; the braiding and anastomosing index decreased to 1 in 1872)", after "(Fig. 4-b...)".

At the end of the title of Figure 9, we propose to add : "... BAI is a Braiding and Anastomosing Index which corresponds to the mean number of these two types of channels (channels showing stagnant water have been excluded)".
* * *
AR-AC: As asked by Reviewer#1 in RC1, we propose to add the following conclusions summarizing the main findings of our study:

"In this study we show the relevance of considering temporal trajectories in process-based river restoration. An interdisciplinary approach deployed at different spatio-temporal scales has been developed by combining planimetric data with sedimentological, chemical and geochronological analysis, as well as a hydrological model. Prior to anthropogenic disturbances, the hydrosystem was mostly characterized by a high-energy depositional environment of braiding channels with high lateral mobility and important surfaces of gravel bars and pioneer vegetation. Correction works provoked a drastic temporal trajectory change, by intensifying filling of fine and polluted (Zn) sediments in palaeochannels and decreasing flood frequency. In contrast, the floodplain

recorded lower deposition rates by quasi-unpolluted sediments. More recently, canalization resulted in very low sedimentation rates, but strong hydrological and hydrogeological disturbances. Our results highlight potential risks that restoration projects may face and need to mitigate along large rivers, e.g. removal fine and potentially polluted sediments by reactivating erosion/deposition processes in former channels. On the Rohrschollen Island, this risk is reduced by the backwater effect of the agricultural dam which limit lateral erosion in the palaeochannel. On the contrary, floodplain areas outside palaeochannels show thin layers of fine sediments and appear more relevant to restore dynamic lateral channels. Managers may benefit from excavating new channels on such areas, as it has been performed on the Rohrschollen Island. They are even encouraged to develop self-erosion of lateral channels by dynamic floods. Finally, this research underscores the necessity to base functional river restorations on the knowledge of hydrosystem past-trajectories that includes the physico-chemical characterization of sediments in order to maximize restoration efficiency and sustainability."
* * *
AR-AC: In addition, we propose also some other short text modifications (modifications of only one to three words are not listed below):

Study area section, we propose to add: "...from braiding to anastomosing and meandering...", after "Slope decrease and inherited geomorphological factors explain the longitudinal evolution of the channel pattern...".

Study area section, we propose to add: "Before engineering works, it was a braiding and anastomosing fluvial hydrosystem.", after "The Rohrschollen artificial Island is located 8 km South-East of the city of Strasbourg and owes its existence to the construction of a power plant in 1970."

In order to answer to a comment of Reviewer#1 in RC1 concerning the CM diagram method, we propose to add: "... we determined the competence of palaeochannel deposits by plotting the median (D50) and the coarsest percentile (D99) of the grainsize distributions in the CM diagram according to Passega (1964, 1977) and Bravard and Peiry (1999).", after "To further characterize transport and depositional processes, we. . .".

---

## Author Response (AR1)

**Dr. David Eschbach**
Université de Strasbourg
Faculté de Géographie
Laboratoire Image, Ville, Environnement
3, rue de l'Argonne – F – 67083 Strasbourg CEDEX
Email: david.eschbach@live-cnrs.unistra.fr

Strasbourg, 13th March 2018

**Object:** Review of the manuscript HESS-2017-435

Dear Professor Hubert Savenije,

Following up on the feedback given by the Referee#1 in RC1 and RC2, we submit a revised version of our manuscript entitled **"Long-term river trajectories to enhance restoration efficiency and sustainability on large rivers: an interdisciplinary study"** by D. Eschbach, L. Schmitt, G. Imfeld, J.-H. May, S. Payraudeau, F. Preusser, M. Trauerstein and G. Skupinski.

We have carefully considered and implemented the very helpful suggestions made by the referee. Here below, our responses to his comments are marked in blue font for a better readability. In the manuscript, we have marked all our edits in track changes mode.

Again, we would like to sincerely thank you for having given us the opportunity to submit a revised manuscript. We remain at your disposal for any further information that would be required.

Yours sincerely,

David Eschbach,
Corresponding Author (CA),
on behalf of all co-authors.

\*\*\*\*\*\*\*\*\*\*\*\*\*\*\*\*\*\*\*\*\*\*\*\*\*\*

**Referee #1: RC1 (11.10.2017)**

**CA:** Please note that the title was modified by *"Long-term river trajectories to enhance restoration efficiency and sustainability on large rivers: an interdisciplinary study"*

**GENERAL COMMENTS**

**RC1:** The study deals with the long-term evolution of a reach of the Rhine River which underwent some restoration activities. Overall, I think it is a very good work: the novelty is combining reconstruction of morphological evolution with other aspects, specifically with geochemical characteristics of sediments. Some revisions are needed to make some parts more effective, especially the last section (see "specific comments"), and to put the work in a wider context (it

could be useful to summarize one or two key points that comes out from this study and this restoration project).

**CA:** We added the following conclusions summarizing the main findings of our study. Please see p.18, lines 25-43 in the revised version where the following section was added:

*"In this study we show the relevance of considering temporal trajectories in process-based river restoration. An interdisciplinary approach deployed at different spatio-temporal scales has been developed by combining planimetric data with sedimentological, chemical and geochronological analysis, as well as a hydrological model. Prior to anthropogenic disturbances, the hydrosystem was mostly characterized by a high-energy depositional environment of braiding channels with high lateral mobility and important surfaces of gravel bars and pioneer vegetation. Correction works provoked a drastic temporal trajectory change, by intensifying filling of fine and polluted (Zn) sediments in palaeochannels and decreasing flood frequency, though some intense floods occurred. In contrast, the floodplain recorded lower deposition rates by quasi-unpolluted sediments. More recently, canalization resulted in very low sedimentation rates, but strong hydrological and hydrogeological disturbances.*

*Our results highlight potential risks that restoration projects may face and need to mitigate along large rivers, e.g. removal fine and potentially polluted sediments by reactivating erosion/deposition processes in former channels. On the Rohrschollen Island, this risk is reduced by the backwater effect of the agricultural dam which limit lateral erosion in the palaeochannel. On the contrary, floodplain areas outside palaeochannels show thin layers of fine sediments and appear more relevant to restore dynamic lateral channels. Managers may benefit from excavating new channels on such areas, as it has been performed on the Rohrschollen Island. They are even encouraged to develop self-erosion of lateral channels by dynamic floods.*

*Finally, this research underscores the necessity to base functional river restorations on an interdisciplinary knowledge of hydrosystem past-trajectories to maximize restoration efficiency and sustainability. On a practical level, we recommend managers to conduct such studies in geomorphological restoration projects, even if they are less detailed than our study presented in this article, for example in the case of financial constraints. They should at least be based on (i) planimetric analysis (old maps and photographs), (ii) sedimentological prospection (hand auger) combined to both a LIDAR DEM analysis and a rapid study of former large floods and (iii) a physico-chemical analysis of sediments, especially these filling palaeo-channels.*

**SPECIFIC COMMENTS**

**RC1:** "Study area" section (pages 4 and 5). The part dealing with the restoration project could be improved. I think it could be useful to describe a little bit more in detail the restoration project and, in particular, the aims of the project. This would be very helpful for improving the last section of the manuscript (4.4) (see one of the following comments).

**CA:** This was done. Please see below the correction proposed in response to RC2.

**RC1:** Page 8, L. 2. A brief explanation of the CM diagram method would be useful.

**CA:** Please see p.6, lines 13-14 in the revised version. We added:

*"… we characterized transport and depositional processes by plotting the median (D50) and the coarsest percentile (D$_{99}$) of the grain-size distributions in the CM diagram according to Passega (1964, 1977) and Bravard and Peiry (1999)."*

**RC1:** IRSL dating. I have some concerns about using this dating method within this study: is this method appropriate to the temporal scale considered in this study?

**CA:** Indeed, the presented IRSL ages are close to the upper dating range of the method, but it has been shown in previous papers that ages of a few years can reliably be determined using

luminescence methods (e.g. Ballarini et al. 2003, Quat. Geochron.; Madsen et al. 2005, Marine Geol.).

**RC1:** How much reliable are the results?

**CA:** This question applies in general to geochronological data, actually to any kind of data collected. For the present samples, we refer to the detailed discussion in Preusser et al. (2016, Geochronometria). Please, see also below our detailed answer to RC2.

**RC1:** I am specifically referring to Figure 7, which shows that dates have significant errors and reverse ages can be obtained (see pit 2, where there is a reverse relation between sediment depth and age).

**CA:** We cannot follow the reviewer here. What are 'significant errors'? The ages are associated with uncertainties of ca. 10 %, i.e. lower than those related to radiocarbon dating in this time range due to calibration uncertainties. As a matter of fact, there are no reverse ages in the study, see figure 7. The ages are all in excellent agreement within uncertainties.

An additional detailed answer has also be given to RC2 (please see below).

**RC1:** Overall the contribution of IRSL may be considered useful for this study, since it constrains the age of fine sediment deposition, but it would be useful if authors would add some comments on such data. For instance, could alternative dating method be used in a similar context?

**CA:** Again, we refer to the paper by Preusser et al. (2016), providing a full overview of the topic and discussing the issues raised by the reviewer. Since such a discussion is well beyond the scope of the present study, we prefer not to repeat it here. In summary, there is no alternative dating approach.

An additional detailed answer has also be given to RC2 (please see below).

**RC1:** Section 4.4. This is part could be improved: considering the amount and quality of data, I think that the authors could make some efforts to make this part more effective. I think that they should try to go more in detail about the effects of the restoration project. For instance: were the project aims appropriate for this river reach?

**CA:** The effects of the restoration project cannot be developed here. The aim of this section is to highlight how long-term trajectory of the hydrosystem allows identifying the driving factors, amplitude and response-time of past disturbances. We show that this study is unavoidable and contributes to the construction of a restoration project. The effects of the restoration linked with the inherited morphological characteristics are developed in another paper submitted in Geomorphology. However, we will improve the text to highlight the legitimacy (efficiency and sustainability) of the restoration project, which is strengthened by considering the historical context.

An additional answer, much more detailed, has be given to RC2 (please see below).

**RC1:** To what extent are (or will be) those aims achieved?

**CA:** Again, this is developed in the paper submitted in Geomorphology. A retrospective analysis is carried out to determine the efficiency of the restoration project in the basis of a fine monitoring which is leading in short term (3 years).

An additional answer, much more detailed, has be given to RC2 (please see below).

**RC1:** Which are the main limitations of a restoration project carried out at reach scale, such as the one described in this study?
**CA:** Main limitations are developed in section 4.4.2. For instance, we showed that fine sediments could be remobilized as well as pollutants bound in such sediments.
An additional answer has also be given in the responses to RC2 (please see below).

**RC1:** Other examples to improve this section: "in part, this functioning has been targeted by recent restoration efforts" (Page 21, L. 11-12), this could be illustrated more in detail.
**CA:** By "functioning" we mean "the functioning before major engineering works", and by "restoration efforts", we mean the creation of hydromorphological dynamics. Aims of the restoration are twofold: reinstalling lateral and vertical dynamics into the channel and stimulating bedload dynamics and groundwater - surface water exchanges.

We propose to add the following sentence. Please see p.16, lines 16-18 in the revised version:
*"In part, this functioning and processes have been targeted by recent restoration efforts. More specifically, the restoration aims have been to recover bedload transport, lateral and vertical dynamics, as well as groundwater - surface water exchanges".*

**RC1:** "this highlight the impact: "works are irreversible" (Page 21, L. 20-21), this statement requires further explanations.
**CA:** We added the following text p.17, lines 8-10 in the revised version:
*"This highlight the fact that impacts of correction work and further engineering works are irreversible because the removal of very large amounts of fine sediments seems unthinkable. Furthermore, the strong hydrological alteration by the canalization works makes the functional alteration of the hydrosystem irreversible as well."*

Following this idea, we added the following text p.18, lines 8-9:
*"... and/or natural fine sediment removal by the recovery of active bank erosion in lateral channels".*

**RC1:** I am wondering if it could be useful to add a final section (e.g. "Conclusions" or "Final remarks") where major outcomes of this study (both specific and general) could be summarized.
**CA:** This was done. Please see the first comment of RC1 and p.18, lines 25-44 in the revised version.

**RC1:** Page 22, L. 30. This sentence is not clear: I think it would be useful to explain better what could be likely the future evolution of this reach, and I would avoid a direct reference to Lane balance (it is a concept well known among geomorphologists but, probably, not for readers with different backgrounds).
**CA:** This was done. We added the following text p.18, lines 16-17 in the revised version:
*"For example, it will probably be necessary to balance a relative sediment deficit in the upstream section of the new channel by artificial gravel augmentations, in the next years/decades (Eschbach et al., in review)."*

**RC1:** Figure 9. I have some concerns about this figure. Is it really meaningful to calculate sinuosity if channel configuration was multi-thread from 1743 to 1838?
**CA:** In braiding systems, sinuosity can be calculated by using the thalweg of the main channel.

An additional answer, much more detailed, has be given to RC2 (please see below) and the manuscript and figure 9 have been modified accordingly (calculation of a *Braiding and Anastomosing Index*).

**RC1:** Sinuosity is a key characteristic in single-thread channel, while less relevant in multi-thread channel. I think that it is not correct to assume that sinuosity in 1872 was 1 (it does not look like a straight channel!). I am wondering if this figure could be removed.
**CA:** The average axis of the active band (which corresponds roughly here to the 1872 channel) can be used as a reference to calculate the sinuosity, as this was also shown by Malavoi and Bravard (2010). Moreover, this figure cannot be removed because it summarizes the morphological planimetric evolutions driven by the engineering works, including the sinuosity.
An additional answer, much more detailed, has be given to RC2 (please see below; a *Braiding and Anastomosing Index* has been calculated; Figure 9 has also been modified) and the manuscript has been modified accordingly (please see below).

**RC1:** Some suggests concerning terminology: "channelization" or "channelization work" instead of "correction";
**CA:** "Correction" or "correction works" are the specific terminology used in engineering reports, books or articles to consider the first engineering works in the Upper Rhine River (project of Tulla). That explains why we preferred this terminology as "channelization". Moreover, this is previously specified in the introduction.

**RC1:** Page 14, L. 7. "Central bar" instead of "median bar".
**CA:** This has been done.

**TECHNICAL CORRECTIONS**
**RC1:** Page 1 – L. 21. "IRSL" instead of "IRLS"
**CA:** This has been done.

**RC1:** Page 2 – L. 1. It could be better to use a chronological order where several works are cited. Please consider this comment throughout the manuscript.
**CA:** This has been done.

**RC1:** Page 4 – L. 15. Figure 3 (as well as Figure 4, page 5 L. 7) is cited within the main text before Figure 2.
**CA:** We removed the references to these figures.

**RC1:** Page 20, L. 2. What is the meaning of "NN"? Above sea level?
**CA:** NormalNull is a specific altimetric system reference used in the Upper Rhine. This means that the sea level reference is located in Germany (North Sea) and not in France (Mediterranean Sea).

**RC1:** Page 21, L. 5. "Different spatio-temporal scales"?
**CA:** We propose to keep this expression because it is commonly used to describe an interlocking of spatial scales and temporal scales.

**RC1:** Page 21, L. 32. Eschbach et al., submitted is missing in the reference list.
**CA:** This has been done.

**RC1:** Page 22, L. 33. "Short" instead of "medium"?
**CA:** This has been done.

**RC1:** Figure 4c. A legend should be add to explain the two symbols of this figure (i.e. anchor points and RMSE errors).
**CA:** In the figure 4c, the legend of the bold and dotted lines refers to the line style used for the vertical axis. This has been specified in the legend of the Figure 4. Please see p.9 in the revised version.

\*\*\*\*\*\*\*\*\*\*\*\*\*\*\*\*\*\*\*\*\*\*\*\*\*\*\*

**Referee #1: RC2 (10.02.2018)**

**RC2:** I am disappointed by some of the author's replies, also considering that my comments were not too difficult to address and aiming to improve quality and clarity of the paper.
**CA:** We thank you sincerely for your helpful remarks aiming to improve the paper, in quality and clarity. We are profoundly sorry that you considered our first corrections not satisfactory. We have carefully considered all your comments (RC1 & RC2) and have done our best to satisfy your and the editors requests. We sincerely hope our corrections and the revised version will be considered positively.

**RC2:** Specifically, I think that the author could pay more attention to the following aspects:
(1) Dating by IRSL. The authors replied that there are not reverse ages: is PIT 2 showing a "normal" relation between depth and ages of sediments (figure 7)? I understand that this could be the only option for dating: on the other hand, I think it could be useful to say that there were no other options.
**CA:** The IRSL ages shown in Figure 7 for PIT 2 are $179 \pm 35$ (at 40 cm), $170 \pm 26$ (58 cm) and $165 \pm 22$ (97 cm) years. First, it is important to note that these ages all overlap well within the given uncertainties. For a set of three ages there are $n * (n-1)$ possibilities (6 in our cause) of how these ages can be arranged. The end members would be 1-2-3 and 3-2-1 (where 3 is the youngest and 1 is the oldest age). We have computed the likelihood for the different possible combinations based on their uncertainties using the following MATLAB code (by courtesy of Prof. S. Hergarten, Freiburg):

```
n = 1000000;
a = randn(3,n);
a(1,:) = 179 + a(1,:) * 35;
a(2,:) = 170 + a(2,:) * 25;
a(3,:) = 165 + a(3,:) * 22;
[~,index] = sort(a);
index = 100*index(1,:)+10*index(2,:)+index(3,:);
h = hist(index,1:1000);
% probabilites of the 6 possible orders in percent
h = 100*h(h~=0)/n
```

For this we received the following results:

| % | 1 2 3 | 1 3 2 | 2 1 3 | 2 3 1 | 3 1 2 | 3 2 1 |
|---|---|---|---|---|---|---|
| % | 11.47 | 15.22 | 10.08 | 22.51 | 15.00 | 25.72 |

This indicates that the observed order of values has a likelihood of more than 10% and, indeed, we are considering this a statistically likely enough case to be regarded as "normal" (from a statistical point of view). In other words, the only information we can deduct from the dates of PIT 2 is that they reveal a rapid phase of sediment deposition - with an absence of a significant correlation between depth and ages -, which corresponds to the phase of correction works of the Rhine at the Rohrschollen site.

Furthermore, in order to follow your suggestion, we added a sentence to specify that the IRSL method is the only option for dating in this context. Please see p.7 lines 8-9 in the revised version: *", … as any other alternative approach is achievable in this context (Preusser et al., 2016)"*

**RC2:** (2) Section 4.4. The authors replied that "… effects of the restoration project cannot be developed here…". If so, why in the "Introduction " they say "…and to assess potential benefits and limits of the restoration" (page 3, line 5) and "…evaluate efficiency and sustainability of the restoration effects…" (page 3, line 13)? I understand that they want to avoid overlapping between this paper and another one submitted to "Geomorphology": in this case, my suggest would be to make some change in the "Introduction" to make the whole work more consistent.

**CA:** We fully understand the remark of RC2 and we thank him for this helpful comment. From a general point of view, the aim of the paper is to show that long-term trajectory of the hydrosystem is useful to improve the efficiency and the sustainability of the Rohrschollen restoration project (and potentially of other river restoration projects). This is presented in the introduction, as well as in the discussion (section 4.4). We discuss the general knowledge produced thanks to the historical study, and which is useful in the restoration context (limits, benefits, efficiency, sustainability), but we don't aim to show the post-restoration changes observed in the Rohrschollen Island with large detail, because we believe that (i) this would be beyond the scope and topic of the paper, and (ii) would make the paper unnecessarily long. Furthermore, these detailed results have been published by Eschbach et al. (2017) and may be published in the near future (Eschbach et al., in review). Nevertheless, we fully agree with RC1 and RC2 that the restoration project and the morphological evolution of the restored channel should be better presented in the Study area section, notably to strengthen the discussion.

Consequently, we added / modified several sentences in the introduction and the discussion, both modifications being linked together.

Please see p.2 lines 37-39 in the revised version. We modified the objectives in order to make them coherent with the discussion ("evolutionary trends" is more general than "evolution"; the objective (v) opens avenues towards other river restorations):
*"…, (iv) deduce post-restoration evolutionary trends and (v) propose operational outlook to improve efficiency and sustainability of Rohrschollen's restoration, and by extension of other river restoration projects (Sear et al., 1994; Grabowski and Gurnell, 2016)".*

And p.4 lines 3-9, we added some details on the restoration and post-restoration adjustments:
*"As the bankfull discharge of the new channel is 20 $m^3.s^{-1}$, flooding in the Island occurs when the discharge exceed this threshold. A three years monitoring showed that bedload transport, active lateral and vertical morphodynamics occur along the new channel (active bank erosion, formation of bars and logjams, enhancement of pool-riffle sequences, increase of groundwater – surface water exchanges...; Eschbach et al., 2017; Eschbach et al., in review), but not along the Bauerngrundwasser which is affected by the hydraulic backwater of the agricultural dam (Eschbach et al., 2017; Eschbach et al., in review; see also the pictures of Fig. 1)."*

In the Discussion, p.16, lines 17-18, we added:
*"More specifically, the restoration induced, in the new channel, the recovery of bedload transport, lateral and vertical dynamics, as well as groundwater - surface water exchanges."*

P.17, lines 9-15, we completed (two more sentences) and modified the end of the section:
*"... because the removal of very large amounts of fine sediments seems unthinkable. Furthermore, the strong hydrological alteration by the canalization works makes the functional alteration of the hydrosystem irreversible as well. In this constrained context, the main challenge of the restoration was to recover processes as dynamic floods (on the whole island) and an active morphodynamic gravel bed channel in a relatively restricted environment (new channel; see also below). On the basis of an environmental monitoring conducted during three years after the end of the restoration works, it appears that these restoration objectives are attained (Eschbach et al., 2017; Eschbach et al., in review) and that the restoration choices were relevant (see also below)."*

P.17, lines 25-26, we added:
*"This demonstrates once again the relevancy of the principles of this restoration."*

P.17, line 39, we added:
*"... as it is the case on the Rohrschollen Island (new channel)."*

P.17, line 42, we added:
*"... (as it has been performed on the Rohrschollen Island),..."*

P.18, lines 9-10, we added:
*"... (which could be made more difficult if sediments are polluted) and/or natural fine sediments removal by the restoration of active bank erosion in lateral channels."*

P.18, lines 16-17, we added:
*"... to balance a relative sediment deficit in the upstream section of the new channel by artificial gravel augmentations, in the next years/decades (Eschbach et al., in review)."*

P.18, lines 22-23, we added:
*"...and modelling, both in the frame of fluvial hydrosystem temporal trajectories."*

**RC2:** (3) Sinuosity (Figure 9). Yes, I agree that sinuosity can be measured in a braided rivers: the point is that if you are analyzing a multithread river (braided, wandering; see figure 4) other indices would be more useful to be taken into account (e.g. braiding index).

**CA:** In order to take into account the remarks RC1 and RC2 of the Reviewer#1, we have calculated the thalweg's sinuosity on the basis of the straight length of the reach (p.16, Fig. 8). So, the sinuosity in 1872 is 1.09 rather than 1.00, which was effectively wrong. Furthermore, we have added a *Braiding and Anastomosing Index* (BAI; table of Fig. 8, p.16) which corresponds to the mean number of these two types of channels (channels showing stagnant water have been excluded). Indeed, this index shows in a relevant way the modifications of the channel pattern (BAI decreased from 7.90 to 1.00).

As a consequence, the following changes in the revised version are also proposed:
P.3, line 14, we added:
*"Before engineering works, it was a braiding and anastomosing fluvial hydrosystem."*

P.8, lines 7-8, we added:
*"The braiding and anastomosing index ranged between 7.9 and 5.4 (Fig. 8)"*

P.11, line 7, we added:
*"… , the braiding and anastomosing index decreased from 5.36 to 2.45"*

P.11, line 10, we added:
*"(…; the braiding and anastomosing index decreased to 1 in 1872)"*

At the end of the title of Figure 9 (p.16 in the revised version), we added:
*"… BAI is a Braiding and Anastomosing Index which corresponds to the mean number of these two types of channels (channels showing stagnant water have been excluded)".*

**CA:** In addition, we propose additional modifications in the revised version (modifications of only one to three words are not listed below):

P.3, line 3, we added:
*"…from braiding to anastomosing and meandering…"*

P.3, line 14, we added:

[revised manuscript text omitted]

The sustainability of functional restoration efforts and their management remains an open question: on which time scale, for which compartment and over which spatial scales should they be considered? In this context, it appears crucial to continue the post-restoration monitoring over medium short (3-5 years) and long median (> 5-10 years) time scales to evaluate restoration success (Jähnig et al., 2011; Kondolf and Micheli, 1995; Palmer et al., 2005, Jähnig et al., 2011). This opens up avenues for developing integrative methodological approaches to improve pre-restoration knowledge and to implement post-restoration monitoring and modelling, both in the frame of fluvial hydrosystem temporal trajectories.

**5 Conclusions**

In this study we show the relevance of considering temporal trajectories in process-based river restoration. An interdisciplinary approach deployed at different spatio-temporal scales has been developed by combining planimetric data with sedimentological, chemical and geochronological analysis, as well as a hydrological model. Prior to anthropogenic disturbances, the hydrosystem was mostly characterized by a high-energy depositional environment of braiding channels with high lateral mobility and important surfaces of gravel bars and pioneer vegetation. Correction works provoked a drastic temporal trajectory change, by intensifying filling of fine and polluted (Zn) sediments in palaeochannels and decreasing flood

frequency, though some intense floods occurred. In contrast, the floodplain recorded lower deposition rates by quasi-unpolluted sediments. More recently, canalization resulted in very low sedimentation rates, but strong hydrological and hydrogeological disturbances.

Our results highlight potential risks that restoration projects may face and need to mitigate along large rivers, e.g. for example the removal of fine and potentially polluted sediments by reactivating erosion/deposition processes in former channels. On the Rohrschollen Island, this risk is reduced by the backwater effect of the agricultural dam which limits lateral erosion in the palaeochannel. On the contrary, floodplain areas outside palaeochannels show thin layers of fine sediments and appear more relevant to restore dynamic lateral channels. Managers may benefit from excavating new channels on such areas, as it has been performed on the Rohrschollen Island. They are even encouraged to develop self-erosion of lateral channels by dynamic floods.

Finally, this research underscores the necessity to base functional river restorations on the an interdisciplinary knowledge of hydrosystem past-trajectories that includes the physico-chemical characterization of sediments in order to maximize restoration efficiency and sustainability. On a practical level, we recommend managers to conduct such studies in geomorphological restoration projects, even if they are less detailed than our study presented in this article, e.g. for example in the case of financial constraints. They should at least be based on (i) planimetric analysis (old maps and photographs), (ii) sedimentological prospection (hand auger) combined to both a LIDAR DEM analysis and a rapid study of former large floods and (iii) a physico-chemical analysis of sediments, especially these filling palaeo-channels. Therefore, to improve pre-restoration project, we recommend to the manager to conduct, at least, (i) a diachronic historical mapping compared with DEM data, (ii) basic sediment prospection (carried out by hand-augered) and, if possible, (iii) geochemical analysis of the filling material.

[revised manuscript text omitted]